# Incomplete transcripts dominate the *Mycobacterium tuberculosis* transcriptome

Xiangwu Ju[1], Shuqi Li[2], Ruby Froom[2,3], Ling Wang[1], Mirjana Lilic[3], Madeleine Delbeau[3], Elizabeth A. Campbell[3], Jeremy M. Rock[2✉] & Shixin Liu[1✉]

*Mycobacterium tuberculosis* (Mtb) is a bacterial pathogen that causes tuberculosis (TB), an infectious disease that is responsible for major health and economic costs worldwide[1]. Mtb encounters diverse environments during its life cycle and responds to these changes largely by reprogramming its transcriptional output[2]. However, the mechanisms of Mtb transcription and how they are regulated remain poorly understood. Here we use a sequencing method that simultaneously determines both termini of individual RNA molecules in bacterial cells[3] to profile the Mtb transcriptome at high resolution. Unexpectedly, we find that most Mtb transcripts are incomplete, with their 5′ ends aligned at transcription start sites and 3′ ends located 200–500 nucleotides downstream. We show that these short RNAs are mainly associated with paused RNA polymerases (RNAPs) rather than being products of premature termination. We further show that the high propensity of Mtb RNAP to pause early in transcription relies on the binding of the σ-factor. Finally, we show that a translating ribosome promotes transcription elongation, revealing a potential role for transcription–translation coupling in controlling Mtb gene expression. In sum, our findings depict a mycobacterial transcriptome that prominently features incomplete transcripts resulting from RNAP pausing. We propose that the pausing phase constitutes an important transcriptional checkpoint in Mtb that allows the bacterium to adapt to environmental changes and could be exploited for TB therapeutics.

TB is the leading cause of death among infectious diseases[4]. The aetiological agent of TB, Mtb, has an exceptional ability to evade host defence mechanisms and drug treatment[1]. Mtb achieves this feat in part by enacting the appropriate transcriptional output in response to changing environments[2]. Thus, a detailed characterization of the Mtb transcriptome is key to understanding the pathogenesis and persistence of TB. However, many aspects of Mtb transcription remain poorly understood, and previous studies suggest that the operation of the Mtb transcription machinery differs substantially from that of its well-studied *Escherichia coli* (Eco) counterpart[5,6].

Recently, we developed SEnd-seq (simultaneous 5′ and 3′ end sequencing) that enables full-length RNA profiling in a bacterial transcriptome[3]. This method provides a greater resolution of transcript boundaries than standard RNA-seq and has generated new insights into the mechanism of Eco transcription in our previous study[3]. In the current work, we applied SEnd-seq in combination with genetic manipulation and in vitro biochemistry to characterize the Mtb transcriptome and study transcriptional regulation in Mtb.

### Profiling the Mtb transcriptome by SEnd-seq

Compared to Eco, Mtb has a much slower growth rate and is more difficult to lyse. We developed a SEnd-seq method customized for mycobacteria (Extended Data Fig. 1a–c and Methods), which captured the correlated 5′- and 3′-end sequences of individual Mtb transcripts (Fig. 1a,b and Extended Data Fig. 1d,e). Using a SEnd-seq protocol that specifically enriches for 5′-triphosphorylated primary RNAs, we identified 8,873 transcriptional start sites (TSSs) in Mtb as compared to 5,038 in the fast-growing model mycobacterium *Mycobacterium smegmatis* (Msm) and 4,358 in Eco (Fig. 1c, Extended Data Fig. 2a,b and Supplementary Table 1). SEnd-seq also identified 747 leaderless TSSs that lack a 5′ untranslated region (UTR), a known feature of the Mtb trancriptome[7,8] (Extended Data Fig. 2c). Our dataset recapitulates most of the previously identified TSSs but also reveals many new sites (Extended Data Fig. 2d,e). Sequence analysis of our TSS dataset as well as the leaderless TSS subset showed the conserved −10 motif (TANNNT) recognized by the housekeeping σ^A-factor[9] (Extended Data Fig. 2f,g), lending support to the validity of the identified sites. In addition, by using a SEnd-seq protocol for profiling total RNAs that contain both 5′-triphosphorylated primary RNAs and 5′-monophosphorylated processed RNAs, we found that Mtb uses a dearth of strong transcription termination sites (TTSs), defined here as sites where a sharp reduction in the total RNA coverage was observed (Fig. 1d, Extended Data Fig. 2h–k and Supplementary Table 2). The paucity of canonical intrinsic terminators in the Mtb genome is consistent with previous results[10,11]. SEnd-seq also revealed pervasive antisense RNAs (asRNAs) in the Mtb transcriptome (Fig. 1e, Extended Data Fig. 2a,l–p and Supplementary Table 3), also consistent with previous work[8]. Our data further show

[1]Laboratory of Nanoscale Biophysics and Biochemistry, The Rockefeller University, New York, NY, USA. [2]Laboratory of Host-Pathogen Biology, The Rockefeller University, New York, NY, USA. [3]Laboratory of Molecular Biophysics, The Rockefeller University, New York, NY, USA. ✉e-mail: rock@rockefeller.edu; shixinliu@rockefeller.edu

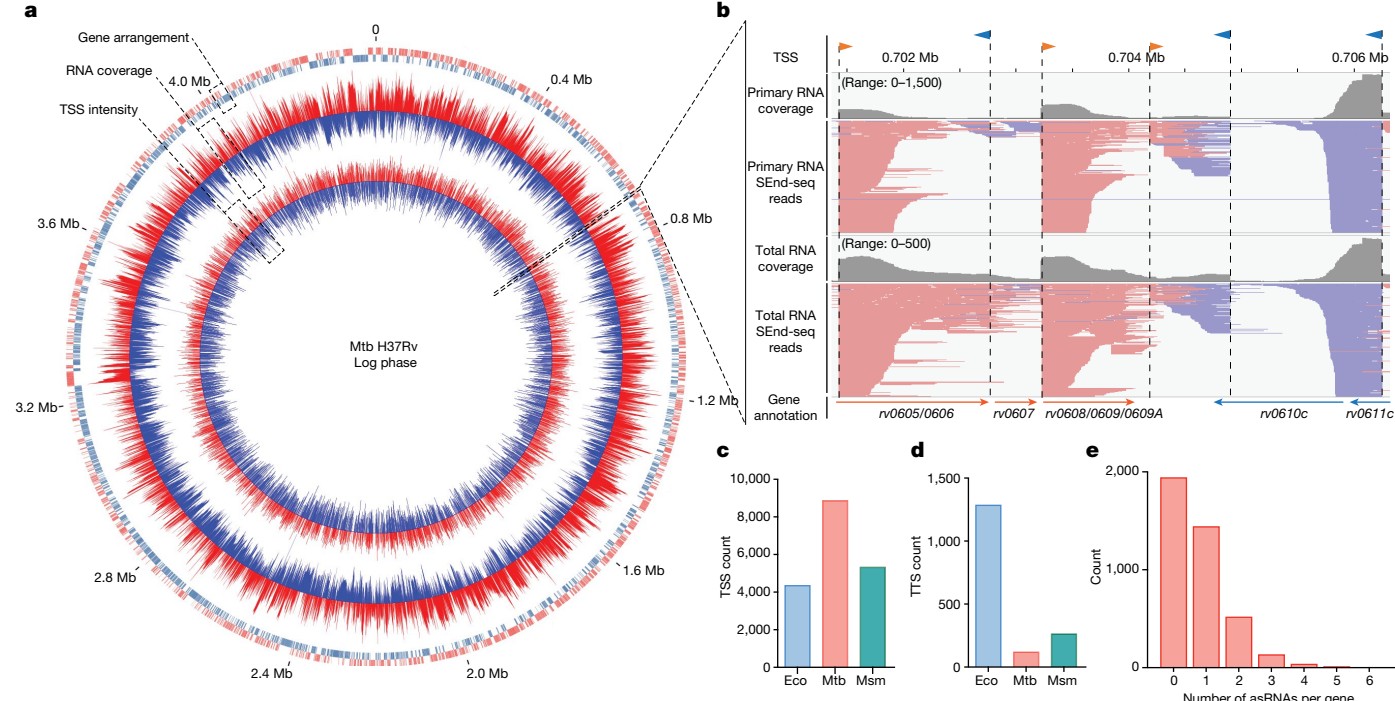

**Fig. 1 | Mtb transcriptome profiling by SEnd-seq. a**, A circos plot showing the transcriptomic profile of log-phase Mtb cells. Outer circle: gene annotation; middle circle: RNA coverage; inner circle: TSS intensity. Red and blue colours represent positive and negative strands, respectively. Mb, megabase.
**b**, A SEnd-seq data track for an example Mtb genomic region showing primary and total RNA coverage (summed over signals from both strands), aligned SEnd-seq reads (red lines: positive-strand transcripts; blue lines: negative-strand transcripts) and TSSs (orange arrows: positive-strand TSSs; blue arrows:

negative-strand TSSs). The primary RNA SEnd-seq data were used for TSS identification. The total RNA SEnd-seq data were used to evaluate the overall RNA level. **c,d**, Bar graphs showing the number of TSSs (**c**) and TTSs (**d**) detected by SEnd-seq for Eco (genome size: 4.6 Mb), Mtb (genome size: 4.4 Mb) and Msm (genome size: 7.0 Mb). The same criteria were used across species for TSS and TTS identification. **e**, Distribution of the number of asRNAs detected by SEnd-seq for each Mtb coding gene.

that the abundance of asRNAs within a given gene is inversely correlated with the corresponding sense transcript level (Extended Data Fig. 2n). Together, these results demonstrate the utility of SEnd-seq in profiling mycobacterial transcriptomes with high resolution and sensitivity.

## The Mtb transcriptome is dominated by short RNAs

A major advantage of SEnd-seq is the precise definition of 5′ and 3′ ends of individual RNA transcripts. By applying this approach, we observed a striking pattern in the RNA coverage in Mtb that differed markedly from what we had observed in Eco[3]. In Eco, the RNA coverage remains largely constant across open reading frames. By contrast, the Mtb RNA coverage drops significantly between 200 and 500 nucleotides (nt) downstream of TSSs (Figs. 1b and 2a,b). The RNA coverage drop-off tends to occur well before the encoded gene's stop codon, yielding a large fraction of incomplete transcripts with heterogeneous 3′ ends. To systematically characterize this RNA drop-off pattern, we annotated 1,930 protein-coding transcription units (TUs) in Mtb, each containing one or several co-directional genes controlled by a major TSS (Extended Data Fig. 3a–d). For each TU, we calculated a numerical 'progression factor' (PF) defined as the ratio of RNA coverage between an upstream zone and a downstream zone using the total (that is, primary and processed) RNA SEnd-seq dataset. Smaller PF values indicate higher fractions of incomplete transcripts (Fig. 2c,d). Indeed, we found that the PF values are predominantly distributed below 1.0, although they do span a wide range of values (Fig. 2e and Supplementary Table 4). TUs with a relatively high or low PF are enriched in genes involved in distinct aspects of Mtb physiology (Extended Data Fig. 3e,f), the implications of which await further investigation. We did not observe a noticeable

dependence of PF on the length of the 5′ UTR (Extended Data Fig. 3g). The RNA coverage drop-off was observed for both log-phase and stationary-phase Mtb cells (Extended Data Fig. 3h). Moreover, the coverage for Mtb asRNAs exhibited a steeper drop-off compared to that for coding TUs (Fig. 2f). Indeed, Mtb asRNAs generally feature a lower PF value than coding TUs (Fig. 2e). We also analysed previous Mtb transcriptomic data obtained by standard RNA-seq[12]—which reported RNA coverage from fragmented short reads—and found a similar drop-off pattern, supporting the findings from our SEnd-seq data (Extended Data Fig. 4a–f).

## Short Mtb transcripts are not due to RNA degradation

We carried out several experiments and analyses to rule out the possibility that the short Mtb transcripts resulted from RNA degradation. First, we found a similar coverage drop-off pattern in both primary RNA and total RNA SEnd-seq datasets (Fig. 1b and Extended Data Fig. 1d,e), ruling out the possibility that the short RNAs are generated by endonucleolytic cleavage, which would yield 5′-monophosphorylated RNAs excluded from the primary RNA sample but preserved in the total RNA sample. Second, we used an anhydrotetracycline (ATc)-inducible expression system to monitor heterologous transcription of the Eco *lacZ* gene in Mtb, which allowed us to track the *lacZ* mRNA length by quantitative PCR (qPCR) as a function of post-induction time. If the short RNAs were mainly due to degradation of full-length transcripts, we would expect to see a burst of RNA signal in the downstream zone at early time points after induction, followed by a decay in signal. However, we observed the accumulation of *lacZ* signal only within 500 nt of the TSS (Extended Data Fig. 4g–i), mirroring the steady-state length profile of endogenous Mtb TUs mapped by SEnd-seq. Last, we used CRISPR

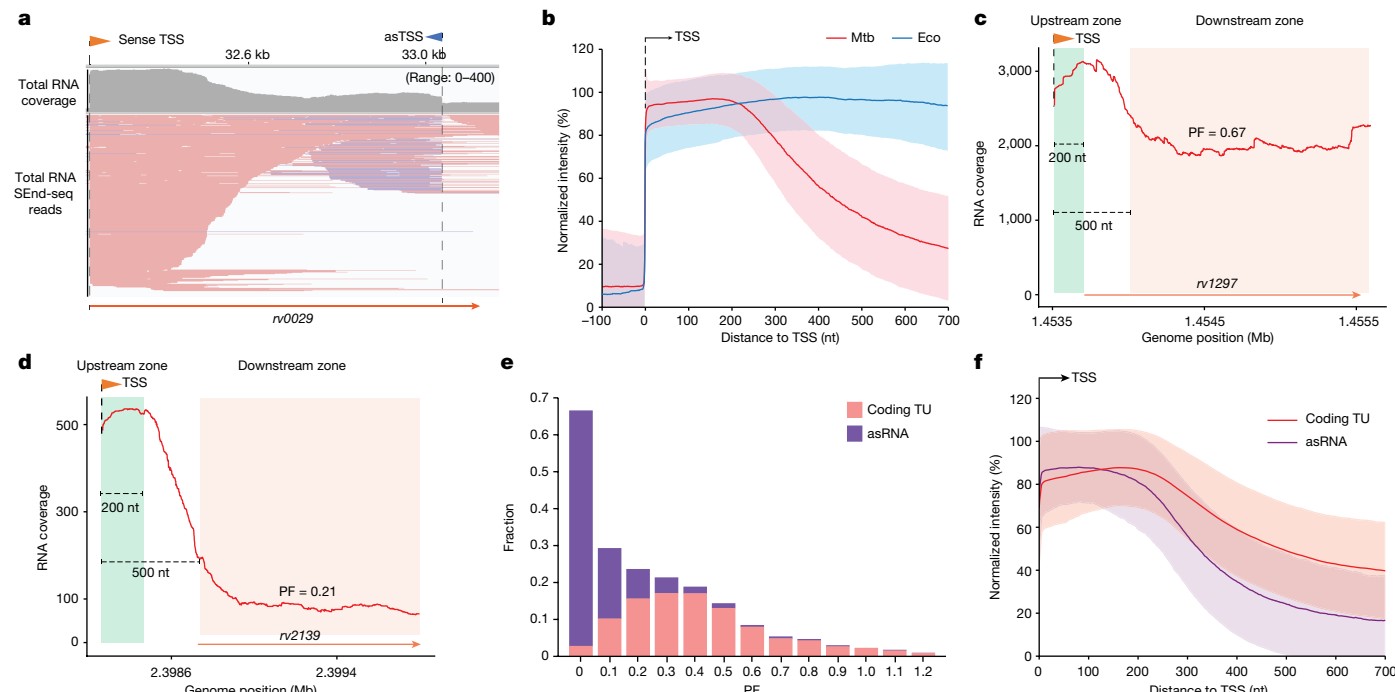

**Fig. 2 | Incomplete RNAs dominate the Mtb transcriptome. a**, A total RNA SEnd-seq data track for an example Mtb TU showing predominantly incomplete RNAs with 5′ ends aligned at the TSS and heterogeneous 3′ ends (red lines: sense transcripts; blue lines: antisense transcripts). kb, kilobase. **b**, Summed SEnd-seq intensities aligned at TSSs for log-phase Mtb (red) and Eco (blue) cells. TSSs without another strong TSS within 700 nt downstream were selected for analysis (*n* = 1,431 for Mtb; *n* = 930 for Eco). The RNA intensities were normalized to the maximum value within a 200-nt window downstream of the corresponding TSS. Coloured lines represent median values and shaded regions represent standard deviations (s.d.). **c**,**d**, SEnd-seq signals for an example Mtb coding TU with a relatively high (**c**) or low (**d**) PF. The upstream and downstream zones for PF calculation are indicated. **e**, Distribution of the PF value for coding TUs (longer than 700 nt) and asRNAs expressed at high levels from log-phase Mtb cells (mean PF = 0.43 for coding TUs; mean PF = 0.09 for asRNAs). Only PF values lower than 1.2 are shown. **f**, Summed SEnd-seq intensities aligned at TSSs for coding TUs (*n* = 1,277) and asRNAs (*n* = 1,440) expressed at high levels. Coloured lines represent median values and shaded regions represent s.d.

interference[13] to individually knockdown RNase or RNase-related genes in Mtb[14] (Extended Data Fig. 5a). After verifying the knockdown efficiency for each strain, we monitored their effects on the PF of select TUs and found that none of the gene knockdowns consistently caused an increase in the PF value (Extended Data Fig. 5b–d). Taken together, these results suggest that the short RNAs prevalently found in the Mtb transcriptome are generated during RNA synthesis rather than during post-transcriptional processing.

## Short RNAs are mostly bound to paused RNAPs

Short transcripts could represent released RNAs due to premature transcription termination[15] or, alternatively, represent nascent RNAs bound to paused transcription elongation complexes. To distinguish between these scenarios, we developed a native elongating transcript SEnd-seq (NET-SEnd-seq) method adapted from a NET-seq protocol developed for Eco[16]. This method uses a His-tagged Mtb RNAP β′-subunit to enrich for RNAP-bound nascent transcripts, which are then subjected to SEnd-seq analysis (Extended Data Fig. 6a–e). Using NET-SEnd-seq, we observed a drop-off pattern in the RNA coverage 200–500 nt downstream of TSSs for both coding TUs and asRNAs, very similar to the drop-off pattern observed in the total RNA SEnd-seq dataset (Fig. 3a,b and Extended Data Fig. 6f–l). This similarity suggests that a large fraction of the short transcripts were associated with RNAPs that paused downstream of the TSS. To further test this interpretation, we carried out chromatin immunoprecipitation followed by high-throughput sequencing (ChIP–seq) experiments for Mtb RNAP using an optimized protocol. Consistent with strong RNAP pausing predicted by NET-SEnd-seq, our ChIP–seq data show RNAP occupancy peaks within

the RNA coverage drop-off region, both for individual TUs (Fig. 3a,c and Extended Data Fig. 7a–f) and in the averaged profile (Fig. 3d). In comparison, we carried out Eco RNAP ChIP–seq experiments and found that Eco RNAP occupancy peaks were located much closer to TSSs (Extended Data Fig. 7g–i), in agreement with previous results[17]. Last, we carried out ChIP–seq experiments for the Mtb σ^A-factor and again observed occupancy peaks within the RNA coverage drop-off region overlapping with the RNAP occupancy peaks (Fig. 3d and Extended Data Fig. 7j–m). Together, our NET-SEnd-seq and ChIP–seq results support a model in which the Mtb RNAP–σ^A holoenzyme has a strong propensity to pause 200–500 nt downstream of the TSS, producing incomplete transcripts. Notably, we did not find strong consensus motifs around the pausing sites (Extended Data Fig. 7n), suggesting that Mtb RNAP pausing in these regions is not sequence specific.

## Rho depletion does not globally lengthen Mtb RNAs

To further support our model that the short transcripts in Mtb are largely associated with paused RNAPs rather than being terminated products, we depleted the transcription termination factor Rho from Mtb cells by CRISPR interference (Extended Data Fig. 8a,b). Consistent with the known essentiality of Rho in Mtb, Rho depletion impaired cell growth and significantly changed the expression level of some essential genes (Extended Data Fig. 8c,d). However, Rho depletion did not significantly alter the PF values for coding TUs as shown by SEnd-seq (Extended Data Fig. 8e), suggesting that the incomplete transcripts were not primarily caused by Rho-mediated premature termination. Instead, we found that *rho* knockdown resulted in a genome-wide increase in the abundance of asRNAs but without a substantial change

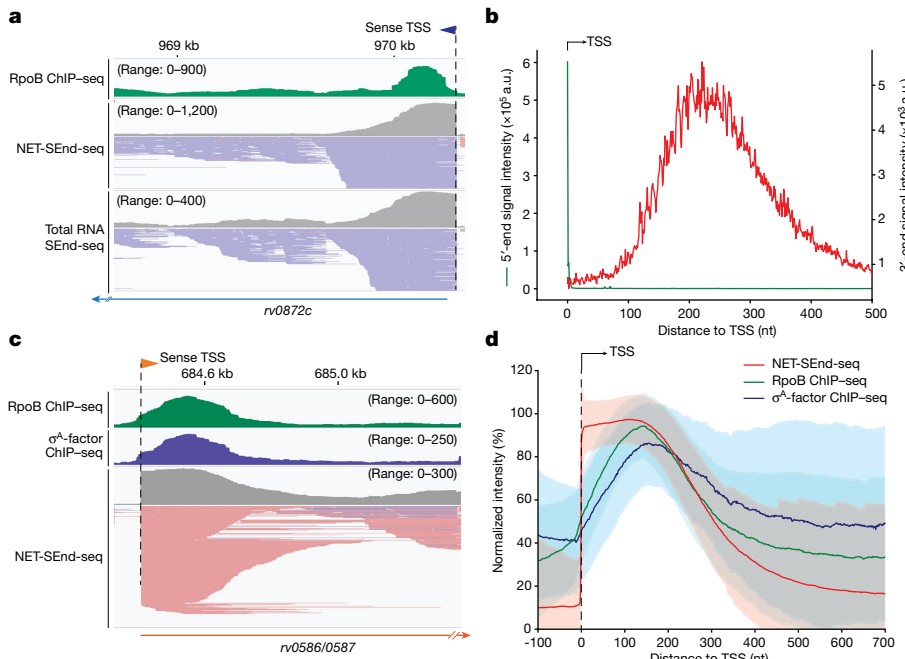

**Fig. 3 | Short RNAs in the Mtb transcriptome are associated with paused elongation complexes. a**, RNAP ChIP–seq, NET-SEnd-seq and total RNA SEnd-seq data tracks for an example Mtb genomic region. **b**, Summed intensities of the 5′-end (green) and 3′-end (red) positions of nascent RNAs from the NET-SEnd-seq dataset ($n = 4,362$). a.u., arbitrary units. **c**, RNAP ChIP–seq, σ[A]-factor ChIP–seq and total RNA SEnd-seq data tracks for an example Mtb genomic region. **d**, Summed intensities of NET-SEnd-seq (red; $n = 1,412$), RNAP ChIP–seq (green; $n = 996$) and σ[A]-factor ChIP–seq (blue; $n = 635$) for Mtb

genomic regions aligned at TSSs. Strong TSSs without another strong TSS within 700 nt downstream were selected for NET-SEnd-seq data analysis. Transcribed regions that exhibited strong RNAP or σ[A]-factor occupancy were selected for ChIP–seq data analysis. The intensities for each position were normalized to the maximum value of the corresponding data track within a 200-nt window downstream of the TSS. Coloured lines represent median values and shaded regions represent s.d.

in their lengths (Extended Data Fig. 8f,g). Our results are consistent with those of a previous RNA-seq study that also reported pervasive transcription in a Rho-depleted Mtb strain[18]. Analysis of the data from both studies showed that the RNA coverage drop-off pattern downstream of TSSs remained largely unchanged following Rho depletion (Extended Data Fig. 8h,i).

## Mtb RNAP–σ[A] synthesizes short transcripts in vitro

The prevalence of short RNAs in Mtb suggests that its transcription machinery has an unexpectedly high probability to enter long-lived pauses well before complete synthesis of full-length transcripts. To determine whether this behaviour is an intrinsic feature of the Mtb RNAP or is mediated by other factors, we carried out in vitro transcription experiments using purified Mtb RNAP–σ[A] holoenzyme and plasmid DNA templates. We used qPCR to analyse the RNA abundances at various locations within the transcribed region as a function of time. In agreement with the in vivo SEnd-seq results, in vitro reactions produced mostly transcripts shorter than 500 nt in length (Fig. 4a and Extended Data Fig. 9a,b). By contrast, the Eco RNAP–σ[70] holoenzyme was able to read through the transcribed region and synthesize full-length transcripts in vitro (Fig. 4b and Extended Data Fig. 9c,d). Next, we added Mtb NusA and NusG factors to the in vitro transcription reaction and found that neither factor substantially enhanced or impaired the synthesis of full-length transcripts by Mtb RNAP–σ[A] (Extended Data Fig. 9e,f).

Given the overlapping Mtb RNAP and σ[A]-factor occupancy observed in vivo from our ChIP–seq experiments, we sought to test whether σ[A]-factor contributes to RNAP pausing in vitro. To this end, we constructed a template containing a preformed DNA bubble that allows RNAP to bypass the need for a σ-factor to initiate transcription. We

found that the Mtb RNAP core enzyme alone was able to synthesize full-length transcripts on this template (Fig. 4c), but the addition of σ[A]-factor restored the short-RNA profile (Fig. 4d). Together, these results indicate that the σ-factor has a crucial role in high propensity of Mtb RNAP to pause within the transcribed region.

## Transcription elongation depends on translation in Mtb

Transcription–translation coupling is an important mechanism for gene regulation in bacteria[19] and has been shown to promote transcription elongation in Eco[20]. We thus posited that active translation may be key to the synthesis of full-length transcripts in Mtb. The steeper drop-off in SEnd-seq coverage for asRNAs than for protein-coding RNAs (Fig. 2f) is consistent with this hypothesis. Moreover, we analysed published Mtb ribosome profiling data[21] and found a positive correlation between the ribosome binding intensity within a coding TU and the corresponding PF value (Fig. 5a,b and Extended Data Fig. 10a). To further examine the impact of translation on transcription elongation in Mtb, we carried out SEnd-seq using Mtb cells treated with the translation inhibitor linezolid[22]. We found that linezolid treatment caused a reduction in RNA coverage in the downstream zone of coding TUs (Fig. 5c and Extended Data Fig. 10e) and significantly decreased the average PF (Fig. 5d), with the notable exception of *whiB7*, which encodes a transcription factor that coordinates a stress response to stalled ribosomes[23,24] (Extended Data Fig. 10b,c). In comparison, rifampicin treatment resulted in a modest increase in the average PF (Fig. 5d). A reduced average PF was also observed for Mtb cells treated with the ribosome-targeting antibiotic clarithromycin[25] (Extended Data Fig. 10d–f) but not for cells treated with streptomycin, which induces ribosome miscoding without preventing translation[26] (Extended Data Fig. 10g–i). Linezolid's negative effect on PF was still observed

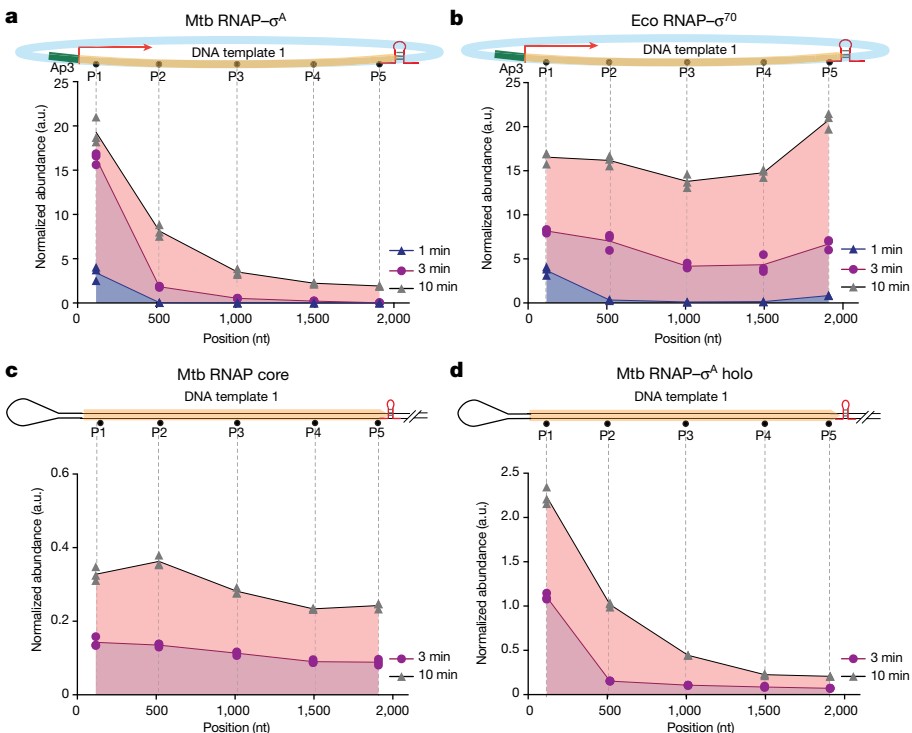

**Fig. 4 | Mtb RNAP–σ^A holoenzyme synthesizes mostly short transcripts in vitro. a**, Length profile of the RNA products from in vitro transcription assays using the Mtb RNAP–σ^A holoenzyme and a plasmid DNA template harbouring an AP3 promoter, a transcribed region (coloured in yellow) and an intrinsic terminator (red hairpin). RNA abundances at different locations of the transcribed region were assessed by qPCR. **b**, Length profile of the RNA products using the Eco RNAP–σ^70 holoenzyme and the same plasmid template as in **a**. Identical reaction conditions were used in **a**,**b**. **c**, Length profile of the RNA products using the Mtb RNAP core enzyme (without σ^A-factor) and a linear DNA template containing a preformed transcription bubble and the same transcribed region and terminator as in the plasmid template in **a**. **d**, Length profile of the RNA products using the Mtb RNAP–σ^A holoenzyme and the same template as in **c**. Data are from three independent measurements.

in Rho-depleted cells (Fig. 5d), reinforcing our conclusion that Rho is not chiefly responsible for the pervasive short RNAs found in Mtb. By contrast, linezolid had little impact on the RNA coverage for asRNAs, consistent with the fact that these transcripts are not efficiently translated (Extended Data Fig. 10j–l).

We further used the inducible *lacZ* system to probe the response of transcription kinetics to translational perturbation. We treated Mtb cells with linezolid immediately before inducing *lacZ* transcription with ATc and found that the RNA abundance in the downstream zone of the *lacZ* gene body was substantially reduced compared to that from cells treated with dimethylsulfoxide (DMSO; Fig. 5e). In addition, we introduced nonsense mutations to the *lacZ* gene body and observed a reduction in the RNA abundance downstream of the ectopic stop codons (Fig. 5f). Together, these results provide evidence that active translation aids in RNAP elongation and full-length mRNA synthesis in Mtb.

## Discussion

In this study, we used SEnd-seq to profile the Mtb transcriptome with high resolution. This led us to the unexpected discovery of pervasive short transcripts in Mtb. The prevalence of short RNAs has also been reported for the Eco transcriptome, but they were found to be predominantly decay intermediates[27]. By contrast, we showed here that the short RNAs in Mtb are primarily nascent transcripts associated with paused RNAPs (Fig. 5g). We further showed that the binding of the housekeeping σ^A-factor induces Mtb RNAP pausing. σ^70-dependent RNAP pausing was also found to be widespread in Eco[16]. However, there are important distinctions between σ-dependent pausing in Eco and Mtb. First, pausing occurs much closer to TSSs in Eco (10–20 nt) than in Mtb

(200–500 nt). It is generally thought that σ^70-factor is released by Eco RNAP immediately after promoter escape[28], although it can sometimes be retained for longer distances[29,30]. Our results showed that Mtb RNAP and σ^A-factor remain associated until at least 200 nt downstream of TSSs. Secondly, Eco σ^70-factor is known to recognize conserved −10-like sequences to induce pausing[31], whereas Mtb σ^A-factor does not seem to have such sequence specificity on the basis of our results. How the diverse σ-factors in Mtb differentially influence RNAP pausing will be an important subject for future research. For example, NET-SEnd-seq assays using His-tagged σ-factors would yield valuable insights into σ-factor occupancy in paused elongation complexes[16]. In addition, it will be interesting to examine whether co-transcribing RNAPs, which regulate the dynamics of Eco transcription via DNA supercoiling[32], also affect the elongation and pausing behaviour of the Mtb transcription machinery.

The functional relevance of transcription–translation coupling varies among different bacterial species[33–35], and its occurrence in mycobacteria has remained largely unexplored. Our results provide evidence that active translation plays a major role in rescuing Mtb RNAP from the paused state and promoting transcription elongation (Fig. 5g). This may confer a fitness advantage to ensure that cells do not waste energy on producing untranslated full-length transcripts[20,36]. Although transcription–translation coupling occurs in Eco and seems to also occur in Mtb, the molecular mechanisms differ. In Eco, Rho is required for transcriptional polarity by mediating premature termination when RNAP is uncoupled from the ribosome[37]. By contrast, Mtb Rho depletion does not relieve the negative effect on transcription elongation by translational inhibition. Whether other factors are involved in transcription–translation coupling in Mtb awaits further investigation. Notably, the universal transcription factor NusG was recently shown to

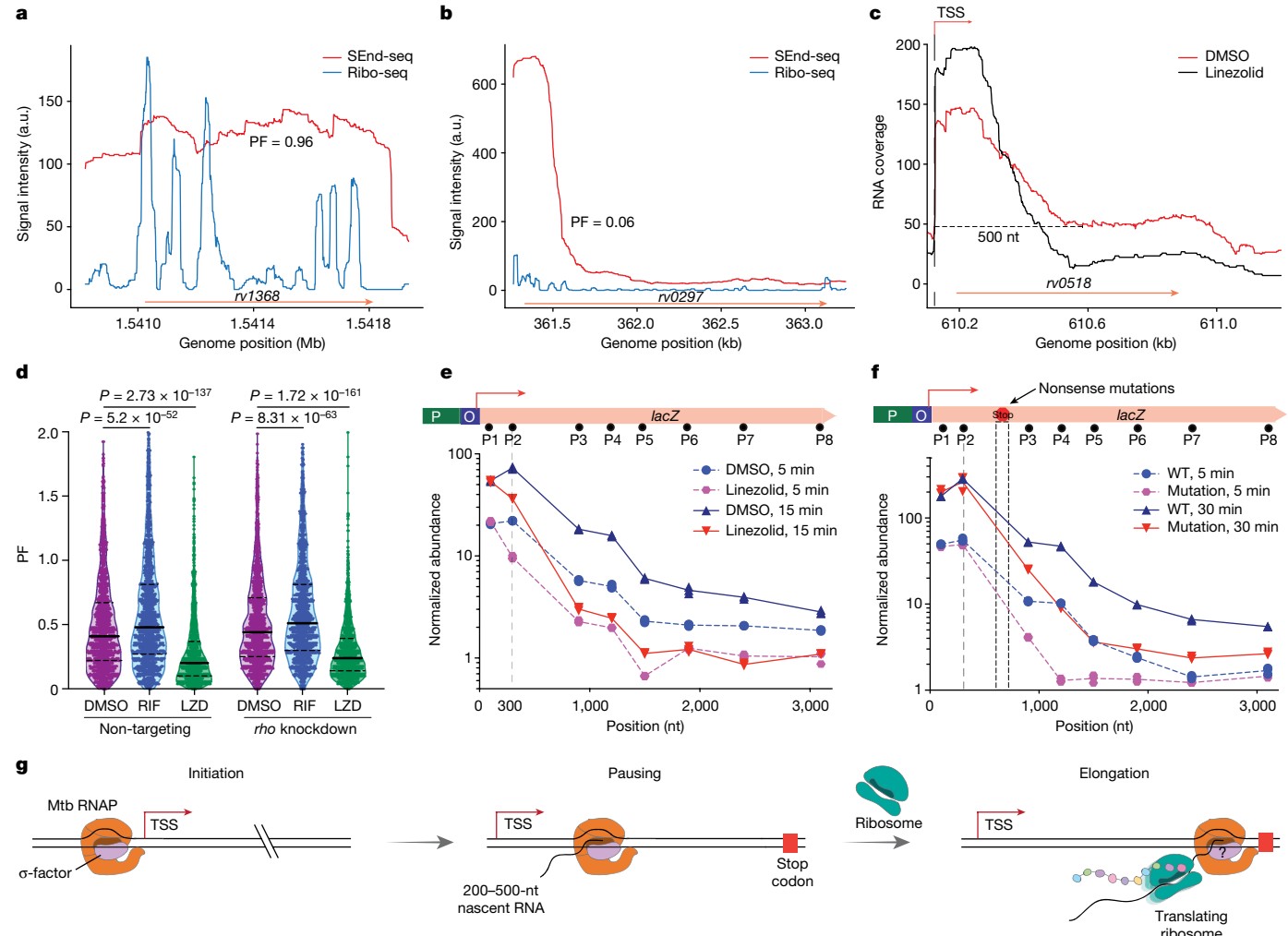

**Fig. 5 | Active translation promotes transcription elongation in Mtb.**
**a**,**b**, SEnd-seq (red) and ribosome profiling (Ribo-seq; blue) signals for a high-PF (**a**) and a low-PF (**b**) coding TU in the Mtb genome. **c**, SEnd-seq signals for an example TU from Mtb cells treated with linezolid (black) and control cells treated with DMSO (red). **d**, A violin plot showing the PF distribution for coding TUs from non-targeting or *rho*-knockdown Mtb cells treated with DMSO, rifampicin (RIF) or linezolid (LZD). TUs expressed at high levels in the DMSO-treated condition were selected for comparison (*n* = 1,380). Solid lines represent the median and dashed lines represent the quartiles. *P* values were determined using two-tailed Student's *t*-test. **e**, Length profile of the RNA products from heterologous *lacZ* transcription in log-phase Mtb cells pre-treated with DMSO or linezolid and measured at 5 min or 15 min after ATc induction. RNA abundances at different positions (P1–P8) were quantified by qPCR and normalized to the DMSO-treated and uninduced condition. P, promoter; O, operator. **f**, Length profile of the RNA products from the wild-type (WT) *lacZ* template or a template containing two nonsense mutations 750 nt downstream of the TSS measured at 5 min or 30 min after ATc induction. RNA abundances were normalized to the uninduced condition for the wild-type template. Data are from three independent measurements. **g**, The working model for Mtb transcription in which the RNAP predominantly pauses 200–500 nt downstream of the TSS and resumes elongation when coupled to a translating ribosome. The binding of the σ-factor contributes to Mtb RNAP pausing, but whether RNAP retains σ-factor throughout the elongation phase remains unclear. Such prevalent transcriptional pausing may allow Mtb cells to integrate regulatory signals and adeptly respond to environmental changes.

be pro-pausing in Mtb in contrast to its anti-pausing role in Eco[38]. In the event of prolonged pausing, the untranslated elongation complexes may eventually undergo premature termination.

The pervasive RNAP pausing downstream of TSSs in Mtb is reminiscent of the promoter-proximal pausing of eukaryotic RNA polymerase II (Pol II), which is a well-established transcriptional checkpoint in metazoans[39]. Undoubtedly, the regulatory factors that induce and release paused polymerases differ across kingdoms. Pol II also tends to pause earlier (within 100 nt downstream of TSSs) than Mtb RNAP. Nevertheless, some of the physiological functions found for Pol II pausing, such as rapid and synchronous activation of gene expression, integration of multiple environmental cues, and coupling of co-transcriptional processes, may be applicable to mycobacterial RNAP pausing. Elucidating the regulatory function of the pervasive transcriptional pausing in Mtb, if any, requires further investigation.

RNAP is a prominent drug target for the antitubercular arsenal[40]. Our study highlights that transcriptional regulation in Mtb departs remarkably from the Eco paradigm. This insight may present unique opportunities for therapeutic intervention. For example, most current RNAP-targeting antibiotics inhibit transcription initiation. It is plausible that paused Mtb RNAP could be targeted for antibiotic development in ways distinct from the mechanism of action for existing antibiotics. Simultaneous targeting of both the ribosome and RNAP may yield synergistic effects on the Mtb transcriptome and proteome. Mtb strains resistant to the key front-line drug rifampicin harbour mutations in RNAP that come with a fitness cost and may affect the pausing behaviour[41], which could make these cells more sensitive to such interventions. Finally, this work serves as a benchmark for dissecting transcriptional reprogramming of Mtb under stress conditions inside the host or following drug treatment.

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

# Methods

## Bacterial strains and growth conditions

The Mtb H37Rv strain (obtained from C. Sassetti) was grown at 37 °C in a minimal medium (Difco Middlebrook 7H9 broth (BD, 271310) supplemented with 0.5% (v/v) glycerol, 0.05% (v/v) tyloxapol (Sigma, T8761), 0.2 g l$^{-1}$ casamino acids (BD, 223050), and 10% (v/v) OADC (oleic acid, albumin, dextrose and catalase; BD, 212351)). The double-auxotrophic Mtb mc$^2$6206 strain (H37Rv $\Delta panCD\Delta leuCD$)[42] (obtained from W. Jacobs Jr) was grown in the minimal medium with an additional 50 mg l$^{-1}$ L-leucine (Sigma, L8000) and 24 mg l$^{-1}$ pantothenic acid (Sigma, P5155). The Msm mc$^2$155 strain (obtained from S. Fortune) was grown in the Middlebrook 7H9 medium supplemented with 0.2% (v/v) glycerol, 0.05% (v/v) Tween-80 (VWR, M126), and 10% (v/v) albumin–dextrose–catalase. Liquid Mtb and Msm cultures were grown at 37 °C in Nalgene sterile square PETG medium bottles with constant agitation. The solid Mtb culture was grown on 7H11 agar (Sigma, M0428) supplemented as described above except for tyloxapol.

## CRISPR interference

Plasmid pIRL58 (Addgene, 166886) bearing the *Streptococcus thermophilus* CRISPR–dCas9 system (dCas9$_{Sthl}$)[13] was used to modulate the RNA expression level of target genes in Mtb mc$^2$6206 cells. Oligonucleotides for single guide RNAs (sgRNAs; Integrated DNA Technologies) were cloned into pIRL58. After verification by Sanger sequencing, pIRL58 and pIRL19 (Addgene, 163634, which supplied the L5 integrase function on a separate suicide vector) were co-transformed into Mtb cells by electroporation using GenePulser (BioRad) at 2,500 V, 700 Ω, and 25 μF. Single colonies were picked from the solid culture plates with 20 μg ml$^{-1}$ kanamycin (Goldbio, K-120) selection after 14–21 days of culture. Target gene knockdown was induced by adding 100 ng ml$^{-1}$ ATc (Sigma, 37919). The sgRNA and primer sequences are listed in Supplementary Table 5.

## SEnd-seq

**RNA isolation.** Bacterial cells were quenched by adding 1× vol of GTC buffer (600 g l$^{-1}$ guanidium thiocyanate, 5 g l$^{-1}$ N-laurylsarcosine, 7.1 g l$^{-1}$ sodium citrate, and 0.7% (v/v) β-mercaptoethanol) to the culture medium immediately before collection and placed at room temperature for 15 min. Cell pellets were collected by centrifugation at 4,000*g* for 10 min at 4 °C, and then thoroughly resuspended in 100 μl TE buffer (10 mM Tris-HCl pH 8.0 and 1 mM EDTA). After the addition of 1 ml TRIzol reagent (Invitrogen, 15596) and 350 mg of glass beads (Sigma, G1145), the cells were immediately lysed in a screw-cap tube by bead beating with the Precellys Evolution homogenizer (Bertin Technologies, 02520-300-RD000) at 10,000 r.p.m. for 4× 45-s cycles with a 60-s interval and chilled with dry ice. After removal of the beads by spinning samples at 12,000 r.p.m. for 5 min at 4 °C, the liquid phase was transferred to a new tube. A 200 μl volume of chloroform was added, and the sample was gently inverted several times until reaching homogeneity. The sample was then incubated for 15 min at room temperature before spinning at 12,000*g* for 10 min at 4 °C. The upper phase (about 600 μl) was gently collected and mixed at a 1:1 ratio with 100% isopropanol. The sample was incubated for 2 h at −20 °C and then centrifuged at 14,000 r.p.m. for 15 min at 4 °C. The pellet was washed twice with 1 ml of 75% (v/v) ethanol, air dried for 5 min and dissolved in nuclease-free water. RNA integrity was assessed with 1% (m/v) agarose gel and the Agilent 2100 Bioanalyzer System (Agilent Technologies, 5067-4626). For antibiotic treatment conditions, Mtb mc$^2$6206 cells were exponentially grown to an optical density at 600 nm (OD$_{600}$) of about 0.8 followed by treatment with a specific antibiotic (30 μg ml$^{-1}$ linezolid (Sigma, PZ0014), 40 μg ml$^{-1}$ clarithromycin (Sigma, C9742), 300 μg ml$^{-1}$ streptomycin (Sigma, S9137), or 50 μg ml$^{-1}$ rifampicin (Sigma, R3501)). At each time point following the treatment, 4 ml of cell culture medium was withdrawn and mixed quickly with 4 ml GTC buffer. The cells were then collected, and the RNA was isolated as described above.

**Library preparation for total RNA SEnd-seq.** A 5 μg quantity of total RNA was mixed with pooled spike-in RNAs used in our previous study[3] at a mass ratio of 300:1 in a total volume of 12 μl. The RNA sample was incubated with a 5′-adaptor ligation mix (1 μl of 100 μM 5′ adaptor (Supplementary Table 5), 0.5 μl of 50 mM ATP, 2 μl DMSO, 5 μl of 50% PEG8000, 1 μl RNase Inhibitor (New England BioLabs, M0314), and 1 μl of High Concentration T4 RNA Ligase 1 (New England BioLabs, M0437)) at 23 °C for 5 h. The sample was then diluted to 40 μl with nuclease-free water and cleaned twice with 1.5× vol of Agencourt RNAClean XP beads (Beckman Coulter, A63987). Immediately following the 5′ adaptor ligation, the eluted RNA was ligated to the 3′ adaptor (Supplementary Table 5) using the same procedure. After incubation at 23 °C for 5 h, the reaction was diluted to 40 μl with water and cleaned twice with 1.5× vol of Agencourt RNAClean XP beads to remove excess adaptors. The sample was subsequently eluted with 0.1× TE buffer and subjected to ribosomal RNA removal with RiboMinus Transcriptome Isolation Kit (Thermo Fisher, K155004) following the manufacturer's instructions. After recovery by ethanol precipitation, the RNA was reverse transcribed to cDNA with *Eubacterium rectale* maturase (recombinantly purified from Eco, obtained from A. M. Pyle)[43] and 5′-phosphorylated and biotinylated reverse transcription primer (Supplementary Table 5). After purification, the cDNA was circularized with TS2126 RNA ligase[44] (obtained from K. Ryan). Double-stranded DNA was synthesized by DNA PolI (New England BioLabs, M0209S). After enzyme inactivation and DNA purification with 1.5× vol of AMPure beads (Beckman Coulter, A63882), the DNA was subsequently fragmented by dsDNA Fragmentase (New England BioLabs, M0348S) at 37 °C for 15 min. The reaction was stopped by adding 5 μl of 0.5 M EDTA and incubated at 65 °C for 15 min in the presence of 50 mM dithiothreitol (DTT). Next, the DNA was diluted to 40 μl with TE buffer and purified with 1.5× vol of AMPure beads. The eluted DNA was used for sequencing library preparation with NEBNext Ultra II DNA Library Prep Kit (New England BioLabs, E7645). Biotinylated DNA fragments were enriched by 5 μl of Dynabeads M-280 Streptavidin (Thermo Fisher, 11205D) and further amplified for 12 cycles by PCR.

**Library preparation for primary RNA SEnd-seq.** A 5 μg quantity of total RNA was used for primary transcript enrichment with our previously published method[3]. In brief, the 5′-triphosphorylated RNA species was specifically capped with 3′-desthiobiotin-GTP (New England BioLabs, N0761) by the Vaccinia Capping System (New England BioLabs, M2080S). The RNA was subjected to 3′ adaptor ligation using the same procedure as described above and subsequently enriched with Hydrophilic Streptavidin Magnetic Beads (New England BioLabs, S1421). After washing thoroughly, the RNA was eluted and reverse transcribed to cDNA as described above. The remaining steps were the same as those for library preparation for total RNA SEnd-seq, except that the DNA library was amplified for 15 cycles.

**Illumina sequencing.** Following PCR amplification, each amplicon was cleaned by 1× vol of AMPure XP beads twice and quantified with a Qubit 2.0 fluorometer (Invitrogen). The amplicon size and purity were further evaluated on an Agilent 2200 Tape Station (Agilent Technologies, 5067-5576). Equal amounts of amplicon were then multiplexed and sequenced with 2 × 150 cycles on an Illumina NextSeq500 or NovaSeq6000 platform (Rockefeller University Genomics Resource Center).

## SEnd-seq data analysis

**Data processing.** After quality filtering and Illumina sequencing adaptor trimming with FASTX-Toolkit (v0.0.13), the raw paired-end reads were merged to single-end reads by using FLASh software (v1.2.11).

The correlated 5′-end and 3′-end sequences were extracted by the custom script (fasta_to_paired.sh) using the SeqKit (v2.4.0) and Cutadapt (v4.1) packages. The inferred full-length reads were generated by Bedtools (v2.31.0) and Samtools (v1.17) after mapping to the reference genome (NC_000913.3 for Eco, NC_008596.1 for Msm and NC_018143.2 for Mtb) with Bowtie 2 (v2.5.1). The full-length reads with an insert length greater than 10,000 nt were discarded. The mapping results were visualized using the IGV genome viewer (v2.4.10). Data analysis and visualization scripts used Python packages including Matplotlib (v3.7.1), Numpy (v1.24.3), Scipy (v1.10.1), bioinfokit (v0.3), and pyCircos (v0.3.0).

**RNA coverage.** Each full-length read was first mapped to the genome in a specific direction. Directional RNA coverage was quantified by summing the number of aligned reads at each mapped nucleotide position. When comparing RNA coverage between samples, data were normalized by the total non-ribosomal RNA amount in each sample. For the samples treated with translation inhibitors, the abundance of spike-in RNAs was used for normalization. Coding TUs and asRNAs with high levels of expression were defined as those with an average RNA coverage of at least 10 for the first 100 nt downstream of the TSS. The circos plot was generated using the Python package pyCircos (github.com/ponnhide/pyCircos, version 0.2.0). The RNA coverage plots were generated using Matplotlib package[45] and custom Python scripts.

**TSS identification.** TSSs were identified from the primary RNA SEnd-seq data using a custom Python script. Only positions with more than 10 reads starting at that position, and with an increase of at least 50% in read coverage compared to its upstream neighbouring position (for example, 50 reads at position −1 and 150 reads at position 0), were retained. Candidate TSS positions within 5 nt in the same orientation were grouped together, and the position with the largest amount of read increase was used as the representative TSS position. Motif analysis around the TSS regions (−40 nt to +5 nt) was carried out by MEME (v5.5.2)[46].

**TTS identification.** Potential TTSs were identified from the total RNA SEnd-seq data at genomic positions with more than 10 reads ending at that position (outside rRNA genes) and with a reduction of more than 40% in read coverage compared to its upstream neighbouring position (for example, 100 reads at position −1 and 50 reads at position 0).

**TU annotation.** TUs were used in this work to analyse the transcription of coding genes. The genome was first segmented into preliminary TUs that contained annotated genes of the same direction. A preliminary unit was further segmented into multiple units if it contained any internal TSS with a strong activity (>2-fold increase in RNA coverage between downstream and upstream of the site for log-phase cell sample). As such, each TU contains a major TSS (TU start site) and possibly additional minor TSSs (<2-fold increase in RNA coverage). The end site of a TU was set to 10 nt before the start of a following co-directional TU, or the middle position between opposite genes that belong to two convergent TUs. TUs shorter than 700 nt and TUs annotated with only rRNA or tRNA genes were excluded from further analysis.

**Antisense transcript annotation.** asRNAs were called if there existed a strong antisense TSS within a given coding TU or if an opposite-direction TSS was found within the non-annotated 400-nt region downstream of a coding TU. The end site of an asRNA was set to the position where the RNA coverage dropped below 25% of the peak value.

**PF analysis.** Each coding TU was assigned with an upstream zone (from 0 to 200 nt downstream of the TSS) and a downstream zone (from 500 nt downstream of the TSS to the end of the TU). If there was another qualified TSS located within the downstream zone, the region downstream of that TSS was excluded from analysis. The ratio between the average RNA intensity of the downstream zone and that of the upstream zone was calculated as the PF for the corresponding TU. For asRNAs, the upstream and downstream zones were defined as 0–200 nt and 500–700 nt downstream of the TSS, respectively. The lower and upper bounds of PF values were set to be 0.0 and 2.0, respectively.

**Gene ontology analysis.** The Database for Annotation, Visualization, and Integrated Discovery (DAVID; v2023q2; https://david.ncifcrf.gov/)[47] was used to carry out gene ontology analysis for Mtb genes with different PF values. The complete list of genes within each set was uploaded to DAVID under the headings of Cellular Compartment, Biological Process, and Molecular Function. Enriched categories with a P value < 0.05 were presented.

## NET-SEnd-seq

Cell collection, lysis, and elongation complex pulldown protocols were adapted from a published study[16] with modifications. Briefly, an ATc-inducible pIRL58 backbone plasmid bearing Mtb *rpoC*-6×His was transformed into Mtb mc[2] 6206 cells and the genome-integrated expression strain was picked as described above. For each pulldown sample, 55 ml of cell culture was prepared. When the cell culture reached the mid-log phase ($OD_{600}$ = 0.5), 100 ng ml$^{-1}$ ATc or an equivalent volume of solvent methanol was added to the medium, and the cells were cultured for another 12 h. After removing 4 ml of cell culture for total RNA extraction, the remaining cell culture was mixed with an equal volume of frozen 2× crush buffer (20 mM Tris-HCl pH 7.8, 10 mM EDTA, 100 mM NaCl, 1 M urea, 25 mM NaN$_3$, 2 mM β-mercaptoethanol, 10% ethanol, 0.4% NP40, and 1 mM phenylmethylsulfonyl fluoride). The cells were subsequently precipitated by centrifugation at 4,000*g* for 10 min at 4 °C, immediately frozen in liquid nitrogen, and stored at −80 °C for at least 1 day. After thawing on ice, the cells were washed twice with 25 ml of cold PBS pH 7.4 and once with 5 ml of cold lysis buffer (20 mM KOH-HEPES pH 7.9, 50 mM KCl, 0.5 mM DTT, 5 mM CaCl$_2$, 10% glycerol, 0.3 mM MgCl$_2$, and 2.5 mM imidazole). The cells were then resuspended in 2 ml of lysis buffer, transferred to two 2-ml lysing matrix B tubes (MP Biomedicals, 116911050), and immediately lysed by bead beating with the Precellys Evolution homogenizer at 10,000 r.p.m. for 4× 45-s cycles with 60-s interval and chilled with dry ice. After centrifugation at 13,000*g* for 5 min, the supernatant was collected into a new 15-ml RNase-free tube. Each lysing matrix B tube was subjected to an additional round of bead beating with 1 ml of fresh lysis buffer and the supernatants were combined. Next, the collected sample was treated with 1 μl TURBO DNase (Life Technologies, AM2238) and incubated at room temperature for 10 min. After centrifugation at 4,000*g* for 10 min at 4 °C, the supernatant was transferred to a new 15-ml tube and incubated with 40 μl pre-washed Ni-NTA beads (Qiagen, 30230) for 1 h at 4 °C with continuous shaking at 100 r.p.m. After immobilization, the beads were washed four times with 5 ml of wash buffer (20 mM Tris-HCl pH 7.8, 1 M betaine, 5% glycerol, 2 mM β-mercaptoethanol, and 2.5 mM imidazole) and five times with 5 ml of pre-elution buffer (20 mM Tris-HCl pH 7.8, 40 mM KCl, 5% glycerol, 2 mM β-mercaptoethanol, and 2.5 mM imidazole). The immobilized complex was subsequently eluted with 300 μl of the pre-elution buffer containing 0.3 M imidazole. The nucleic acids in the eluates were extracted once with 200 μl phenol/chloroform/isoamyl alcohol (25:24:1, v/v/v) and once with 200 μl chloroform. The top aqueous phase was collected and precipitated by 3× volumes of ethanol, 0.1× vol of 3 M sodium acetate pH 5.2, and 2 μl glycogen (Thermo Fisher, AM9510). After precipitation at −20 °C overnight and maximum-speed centrifugation for 20 min, the pellet was washed twice with 300 μl of 75% ethanol. The pellet was then dissolved in 50 μl nuclease-free water and treated with 0.5 U Turbo DNase at 37 °C for 15 min. The residual

RNA was extracted by phenol/chloroform/isoamyl alcohol, precipitated by ethanol and recovered in 11.5 µl of nuclease-free water. A 1 µl volume of spike-in RNA was added to each RNA sample, and the RNAs were ligated to a 3′ adaptor. The remaining steps were the same as those described above for total RNA SEnd-seq. The DNA library was amplified for 16 cycles by PCR.

## ChIP–seq
A 50 ml volume of mid-log phase Mtb cells ($OD_{600}$ = 0.8–1.0) were treated with 1% formaldehyde while the culture was agitated at room temperature for 30 min. Crosslinking was quenched by adding glycine to a final concentration of 250 mM for another 30 min while stirring at room temperature. The cells were pelleted by centrifugation at 4,000$g$ for 10 min at 4 °C and washed three times with 20 ml of cold PBS and 0.1× protease inhibitor (Sigma, P8465). The cell pellet was stored at −80 °C for at least one day. After thawing on ice, the cells were washed once with 5 ml of IP lysis buffer (20 mM KOH-HEPES pH 7.9, 50 mM KCl, 0.5 mM DTT, 5 mM $CaCl_2$, and 10% glycerol) and resuspended in 2 ml of IP lysis buffer. The cells were then transferred to two 2-ml lysing matrix B tubes (MP Biomedicals, 116911050) and immediately lysed by bead beating with the Precellys Evolution homogenizer at 10,000 r.p.m. for 4× 45-s cycles with a 60-s interval and chilled with dry ice. After centrifugation at 13,000$g$ for 5 min, the supernatant was collected into a new 15-ml RNase-free tube. Each lysing matrix B tube was subjected to an additional round of bead beating after adding 1 ml of fresh IP lysis buffer. After centrifugation at 4,000$g$ for 10 min at 4 °C and sampling for input control, 4 ml of supernatant was transferred to a new 15-ml tube and incubated with 0.75 µl of micrococcal nuclease (New England BioLabs, M0247S) at 37 °C for 15 min with continuous shaking. The reaction was stopped by adding EDTA at a final concentration of 25 mM, and the supernatant was transferred to a new 15-ml tube after centrifugation at 4,000$g$ for 10 min at 4 °C. A 3 µl volume of anti-Eco σ$^{70}$-factor antibody (BioLegend, 663208; 1:1,333 dilution) or 5 µl of anti-Eco RNAP β-subunit antibody (BioLegend, 663903; 1:800 dilution) was used to immunoprecipitate Mtb σ$^A$-factor and Mtb RNAP, respectively. After overnight incubation, 40 µl of pre-washed protein A/G agarose beads (Thermo Fisher, 26159) were added and incubated for 2 h at 4 °C and for another 30 min at room temperature. The beads were then washed ten times with 5 ml IPP150 buffer (10 mM Tris-HCl pH 8.0, 150 mM NaCl, and 0.1% NP40) and once with 5 ml TE buffer. Next, the DNA was eluted with 150 µl of elution buffer (50 mM Tris-HCl pH 8.0, 10 mM EDTA, and 1% SDS) followed by 100 µl TE buffer with 1% SDS. After thoroughly removing the beads by centrifugation at 2,000$g$ for 5 min at 4 °C, the combined supernatants were incubated with 1 mg ml$^{-1}$ Pronase (Sigma, 537088) at 42 °C for 2 h and then at 65 °C for 9 h. The sample was cleaned twice with 200 µl of phenol/chloroform/isoamyl alcohol (25:24:1, v/v/v) and recovered by ethanol precipitation. Finally, the sequencing libraries for immunoprecipitated DNA and input control were prepared using the NEBNext Ultra II DNA Library Prep Kit. After sequencing and quality filtering, the reads were mapped to the Mtb genome using Bowtie 2. The ChIP–seq signals were extracted and plotted using custom Python scripts.

## Analysis of deposited RNA-seq data
The RNA-seq datasets SRR5689224 and SRR5689225 (BioProject PRJNA390669)[12] from log-phase Mtb cells cultured in dextrose-containing medium were used to compare the RNA coverage between SEnd-seq and RNA-seq. The RNA-seq datasets SRR5061507, SRR5061514, SRR5061706 and SRR5061510 (BioProject PRJNA354066)[18] from Mtb cells with Rho depletion were used to compare to the *rho*-knockdown SEnd-seq datasets. The deposited datasets were downloaded from the National Center for Biotechnology Information. After read extraction and quality filtering, the reads were mapped to the Mtb genome using Bowtie 2 (v2.5.1). The RNA intensities were extracted and plotted using custom Python scripts.

## Analysis of deposited Ribo-seq data
Mtb Ribo-seq data were downloaded from the EMBL-EBI database (E-MTAB-8835)[21]. After read extraction and quality filtering, the reads were mapped to the Mtb genome using Bowtie 2. The directional ribosome binding signals were extracted and plotted using a custom Python script.

## Immunoblot
Mtb cells were lysed with TRIzol reagent as described above, and protein samples were extracted following a TRIzol-based protein extraction protocol provided by the manufacturer. Immunoblotting was carried out as described previously[48]. Antibodies against His-tag (Santa Cruz, sc-8036; 1:1,000 dilution), Mtb Rho (obtained from D. Schnappinger; 1:200 dilution), and Eco RpoB (BioLegend, 663903; 1:1,000 dilution) were used.

## qPCR
A 1–10 µg amount of total RNA was treated with 0.5 µl of TURBO DNase (Life Technologies, AM2238) at 37 °C for 30 min to remove the genomic DNA. The sample was diluted to 100 µl with RNase-free water and then cleaned three times with 100 µl of $H_2O$-saturated phenol/chloroform/isoamyl alcohol (25:24:1, v/v/v). After ethanol precipitation, 1 µg of RNA was reverse transcribed to cDNA with the High-Capacity cDNA Reverse Transcription Kit (Thermo Fisher, 4368814) following the manufacturer's instructions. qPCR was conducted using synthesized primers and the SYBR green master mix (Thermo Fisher, 4309155) on a QuantStudio5 Real-Time PCR System (Thermo Fisher). The relative RNA abundance was presented as the signal ratio between the target transcript and the reference 16S rRNA from the same sample using the formula: $2^{Ct(16S) - Ct(target)}$, in which $C_t$ denotes the cycle threshold.

## Inducible *lacZ* transcription in Mtb
Plasmid pIRL58 was modified by removing the sgRNA expression cassette and replacing the dCas9$_{Sth1}$ gene body with the Eco *lacZ* coding region, allowing the synthesis of *lacZ* RNA under the control of ATc-inducible promoter $P_{tet}$. The modified plasmid was co-transformed into Mtb mc$^2$6206 cells with pIRL19 as described above. Cells from a single colony of Mtb $P_{tet}$-*lacZ* after selection were exponentially grown to an $OD_{600}$ of about 0.8 followed by the addition of 100 ng ml$^{-1}$ ATc to induce *lacZ* transcription. After induction, 4 ml of cell culture was withdrawn at indicated time points and mixed with 4 ml GTC buffer in a new tube as sample $t$ (St). One extra sample taken immediately before ATc addition was referred to as S0. After RNA isolation and TURBO DNase treatment as described above, 1 µg of total RNA was used to synthesize the cDNA for qPCR. The relative *lacZ* mRNA abundance at each time point is defined as $2^{Ct(S0) - Ct(St)}$, in which Ct denotes the cycle threshold.

## In vitro transcription
DNA fragments were amplified by PCR from Mtb genomic DNA with primer sets listed in Supplementary Table 5. An AP3 promoter sequence was inserted into one end of the fragment and an intrinsic terminator (derived from TsynB in pIRL58) was placed at the other end. The DNA fragment was then incorporated into the pUC19 plasmid. The plasmid templates were prepared from Eco DH5α cells and subsequently treated with 2 µl RNase A (Thermo Fisher, EN0531) for 30 min and 2 µl Proteinase K (New England BioLabs, P8107S) for 1 h. The plasmid templates were cleaned three times with phenol/chloroform/isoamyl alcohol (25:24:1, v/v/v) and recovered by ethanol precipitation.

To prepare templates with a preformed bubble, the DNA fragment containing the intrinsic terminator was amplified from the plasmid DNA described above by PCR. The product was cleaned with QIAQuick PCR purification kit (Qiagen, 28104) and phenol/chloroform/isoamyl alcohol (25:24:1, v/v/v). The bubble template was constructed by ligating a DNA adaptor (NEBNext adaptor for Illumina) to each end of the

DNA fragment using NEBNext Ultra II DNA Library Prep Kit. After XbaI digestion (cut site immediately after the terminator), the DNA template was purified using AMPure XP beads.

Purified Mtb RNAP, σ[A]-factor, NusA, and NusG were prepared as described previously[38,49,50]. The in vitro transcription mixture contained 2 µl of 10× transcription buffer (200 mM Tris-acetate pH 7.9, 0.5 M potassium acetate, 100 mM magnesium acetate, 10 mM DTT, and 50 µg ml$^{-1}$ BSA), 1 µl RNase inhibitor, 0.5 pmol of DNA template, and 2 pmol of Mtb RNAP holoenzyme (or core RNAP alone) in a 20 µl volume. The mixture was incubated at 37 °C for 15 min before the addition of rNTPs (100 µM each). At indicated time points, the reaction was quenched by adding EDTA at a final concentration of 20 mM and 2 µl of Proteinase K and incubating for 30 min. The reaction was then diluted to 100 µl with RNase-free H$_2$O and cleaned three times with phenol/chloroform/isoamyl alcohol (25:24:1, v/v/v). After ethanol precipitation and resuspension with 30 µl RNase-free H$_2$O, 0.5 µl DNase I (New England BioLabs, M0303S), 3.5 µl of DNase buffer, and 1 µl RNase inhibitor were added. After incubation at 37 °C for 30 min, the RNA product was cleaned three times with phenol/chloroform/isoamyl alcohol (25:24:1, v/v/v) and recovered by ethanol precipitation. Half of the RNA was converted to cDNA with the High-Capacity cDNA Reverse Transcription Kit and evaluated by qPCR as described above. RNA abundances were normalized to a diluted plasmid DNA sample with a concentration of 0.033 ng ml$^{-1}$.

## Statistics

Statistical analyses were conducted with Excel (version 16.178.3) or GraphPad Prism (version 10.1.0). GraphPad Prism (version 10.1.0) or the Python Matplotlib package (version 3.7.1) was used for plotting.

## Reporting summary

Further information on research design is available in the Nature Portfolio Reporting Summary linked to this article.

## Data availability

SEnd-seq, NET-SEnd-seq and ChIP–seq datasets from this study have been deposited in the Gene Expression Omnibus with the accession number GSE211992 (BioProject PRJNA873109). Source data are provided with this paper.

## Code availability

The custom scripts in this study are available at https://github.com/LiuLab-codes/Mtb_transcriptome_profiling.

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

**Acknowledgements** We thank R. Landick, A. M. Pyle and D. Schnappinger for sharing reagents; M. DeJesus, R. Gong, G. Chua and Rockefeller University Genomics Resource Center for technical assistance; and S. Darst for a critical reading of the manuscript. This work was supported by a National Institutes of Health (NIH) grant (5R01GM114450 to E.A.C.), NIH New Innovator Awards (1DP2AI144850-01 to J.M.R.; 1DP2HG010510-01 to S. Liu), the Stavros Niarchos Foundation Institute for Global Infectious Disease Research (to E.A.C., J.M.R. and S. Liu), the Rita Allen Foundation (to J.M.R.), the Robertson Foundation (to S. Liu) and the Alfred P. Sloan Foundation (to S. Liu).

**Author contributions** X.J., J.M.R. and S. Liu conceived the project. X.J. carried out the experiments and data analysis. S. Li and J.M.R. provided resources for mycobacterial sample collection. R.F., L.W., M.L., M.D. and E.A.C. assisted with in vitro transcription experiments. X.J., J.M.R. and S. Liu wrote the manuscript with inputs from S. Li, R.F. and E.A.C.

**Competing interests** The authors declare no competing interests.

**Additional information**
**Correspondence and requests for materials** should be addressed to Jeremy M. Rock or Shixin Liu.

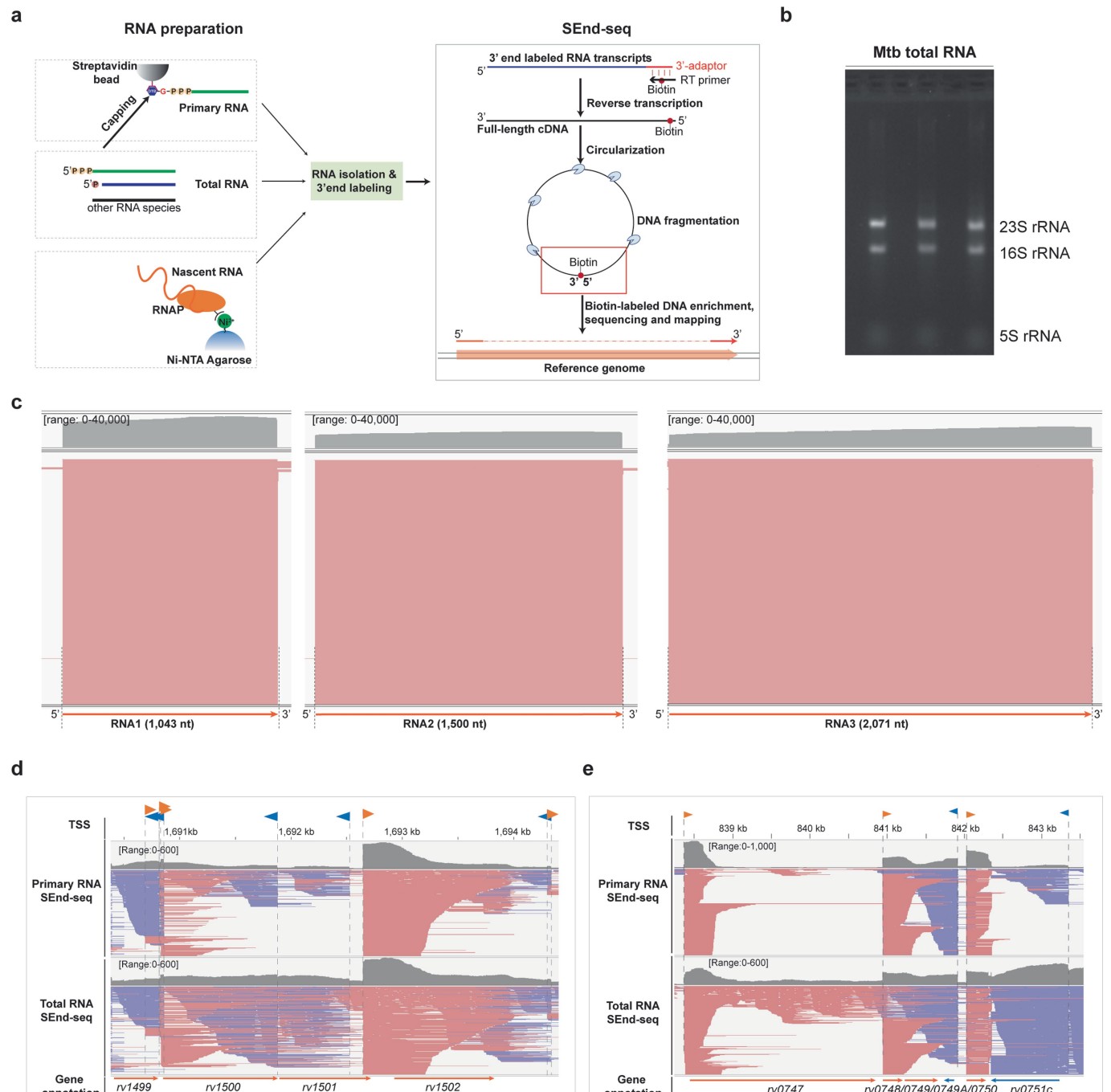

**Extended Data Fig. 1 | Workflow, quality control, and example data tracks of Mtb SEnd-seq. a**, Workflow of Mtb SEnd-seq for primary RNA, total RNA, and nascent RNA associated with elongation complexes. See Methods for details. **b**, Gel analysis for the total RNA isolated from Mtb cells showing the abundant ribosomal RNA species. This experiment was repeated more than three times with similar results. **c**, SEnd-seq data tracks for three spike-in RNAs with different lengths, which were pooled with total cellular RNA before the preparation of sequencing libraries. The uniform coverage of the spike-in RNAs indicates that the library preparation steps had minimal impact on the RNA integrity. **d-e**, Primary RNA and total RNA SEnd-seq data tracks for two example genomic regions from log-phase Mtb cells (red lines: positive-strand transcripts; blue lines: negative-strand transcripts; orange arrows: positive-strand TSSs; blue arrows: negative-strand TSSs).

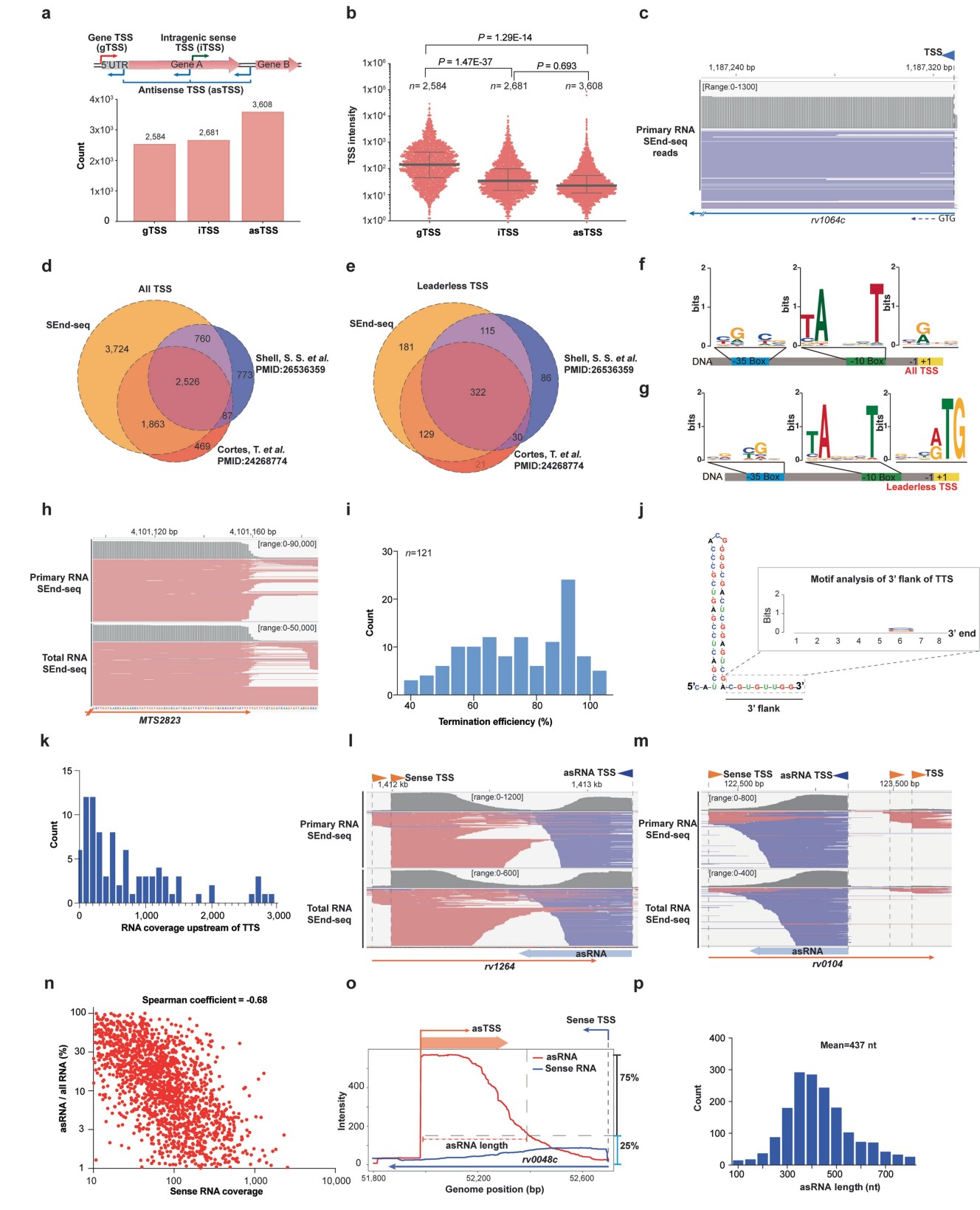

**Extended Data Fig. 2** | See next page for caption.

**Extended Data Fig. 2 | Characterization of Mtb TSSs, TTSs, and asRNAs detected by SEnd-seq. a**, (Top) Schematic showing different categories of TSS based on its location and orientation. (Bottom) Number of gTSSs, iTSSs, and asTSSs in the Mtb genome identified by SEnd-seq. **b**, Distribution of TSS intensities for the gTSSs ($n = 2,584$), iTSSs ($n = 2,681$), and asTSSs ($n = 3,608$) described in **a**. The bars indicate mean values with interquartile range. $P$ values were determined using two-tailed Student's $t$-test. **c**, Primary RNA SEnd-seq data track showing an example leaderless TSS in Mtb. **d-e**, Venn diagram showing the overlap of all TSSs (**d**) or leaderless TSSs (**e**) identified by SEnd-seq in this study with those reported by two previous studies[7,8]. **f-g**, Motif analysis for the +1 site, −10 element, and −35 element for all Mtb TSSs (**f**) and leaderless TSSs (**g**) identified by SEnd-seq. **h**, SEnd-seq data track showing an example TTS in Mtb. **i**, Distribution of the termination efficiencies for the TTSs in the Mtb genome identified by SEnd-seq. The lower bound to qualify for a TTS was set to 40%. **j**, Secondary RNA structure upstream of the example TTS shown in **h**. (Inset) Motif analysis for the 3′ flanking sequences of the RNA hairpins upstream of all identified Mtb TTSs ($n = 121$) showing a lack of conserved motif. **k**, Distribution of the RNA coverage upstream of each TTS identified by SEnd-seq. Only sites with an RNA coverage less than 3,000 are shown here. Note that we cannot detect potential TTSs for very lowly expressed genes due to the read threshold used in our TTS identification criteria (see Methods). **l-m**, Primary RNA and total RNA SEnd-seq data tracks for two example Mtb genomic regions showing an abundance of antisense RNAs (blue lines). **n**, Scatter plot showing the anticorrelation between the percentage of asRNAs in each TU ($n = 1,930$) and the summed coverage of the corresponding coding RNAs. Spearman correlation coefficient is shown. **o**, SEnd-seq signals for an example Mtb asRNA demonstrating the definition of asRNA length used in this study. **p**, Distribution of the length of asRNAs identified by SEnd-seq in log-phase Mtb cells.

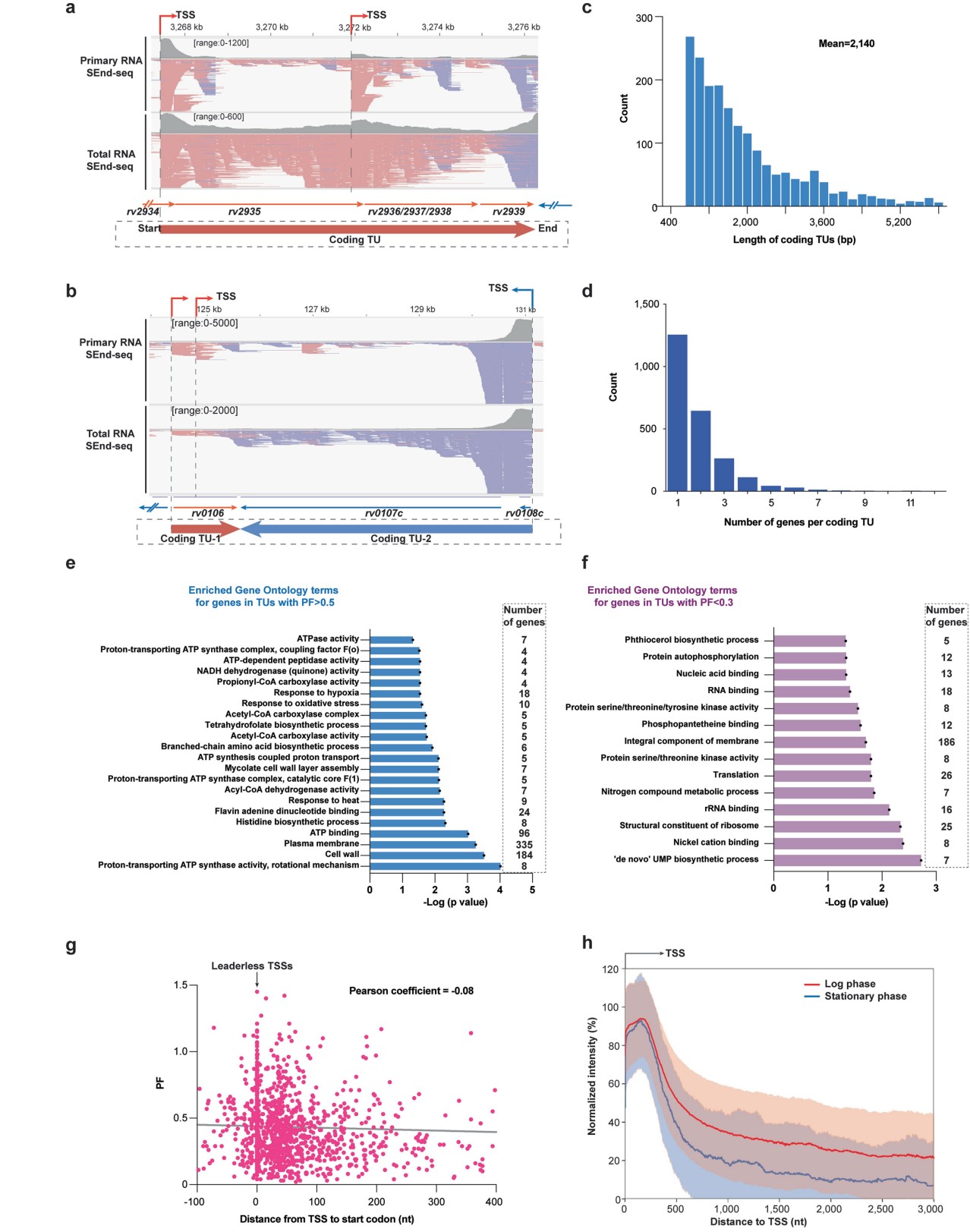

**Extended Data Fig. 3 | Characterization of Mtb coding transcription units (TUs) detected by SEnd-seq. a-b**, Primary and total RNA SEnd-seq data tracks for two example Mtb genomic regions showing a TU consisting of several co-directional genes (**a**) and two convergent TUs (**b**). **c**, Distribution of the length of Mtb coding TUs. TUs shorter than 6,000 nt are shown here. **d**, Distribution of the number of annotated Mtb genes within each coding TU. **e-f**, Gene Ontology analysis for Mtb genes in TUs with a progression factor (PF) > 0.5 (**e**) or with a PF < 0.3 (**f**). **g**, Scatter plot showing the lack of strong correlation between the distance from the major TSS of a coding TU to its start codon (0 indicates a leaderless TSS) and the PF of the same TU. Pearson correlation coefficient is shown. **h**, Summed SEnd-seq intensities aligned at TSSs for log-phase (red; $n = 1,390$) and stationary-phase (blue; $n = 302$) Mtb cells. Colored lines represent median values and shaded regions represent s.d.

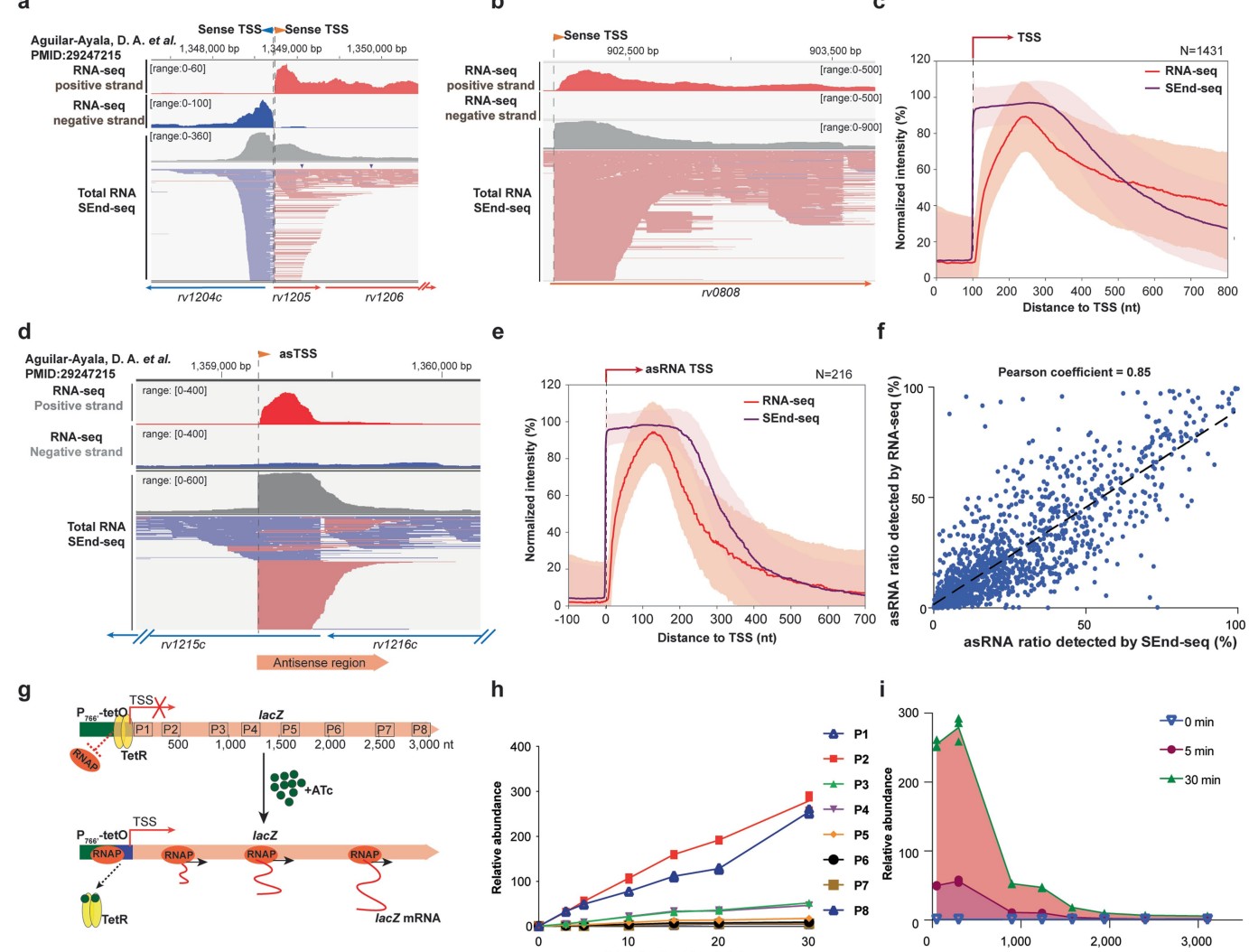

**Extended Data Fig. 4 | Supporting evidence for the pervasive short transcripts in Mtb cells. a-b**, Standard RNA-seq (ref. 12) and SEnd-seq (this study) data tracks for two example TUs in log-phase Mtb cells. As expected based on methodological differences, the boundary features are less pronounced in the RNA-seq dataset than in the SEnd-seq dataset. Nonetheless, the overall expression levels are comparable between the two methods. **c**, Summed intensities of RNA-seq and SEnd-seq reads for Mtb genomic regions aligned at TSSs. Strong TSSs that did not show another strong TSS within a 700-nt downstream window were selected for analysis (*n* = 1,431). Colored lines represent median values and shaded regions represent s.d. **d**, Standard RNA-seq[12] and SEnd-seq data tracks for an example asRNA. **e**, Summed intensities of RNA-seq and SEnd-seq reads aligned at TSSs for highly expressed asRNAs in both datasets (*n* = 216). Colored lines represent median values and shaded regions represent s.d. **f**, Scatter plot showing the correlation between the percentage of asRNAs in selected TU obtained by RNA-seq and the corresponding value obtained by SEnd-seq. TUs that exhibited high expression of either sense or antisense transcripts in both datasets were selected (*n* = 1,826). Pearson correlation coefficient is shown. **g**, Experimental scheme for measuring the kinetics of heterologous *lacZ* transcription in Mtb cells induced by ATc. **h**, qPCR results for the RNA abundances at various locations (P1 to P8) across the *lacZ* gene body measured at different time points after ATc induction. **i**, Length profile of mRNA products from ATc-induced *lacZ* transcription in Mtb measured at different time points after ATc induction. RNA abundances were normalized to the 0-min values for each probe position.

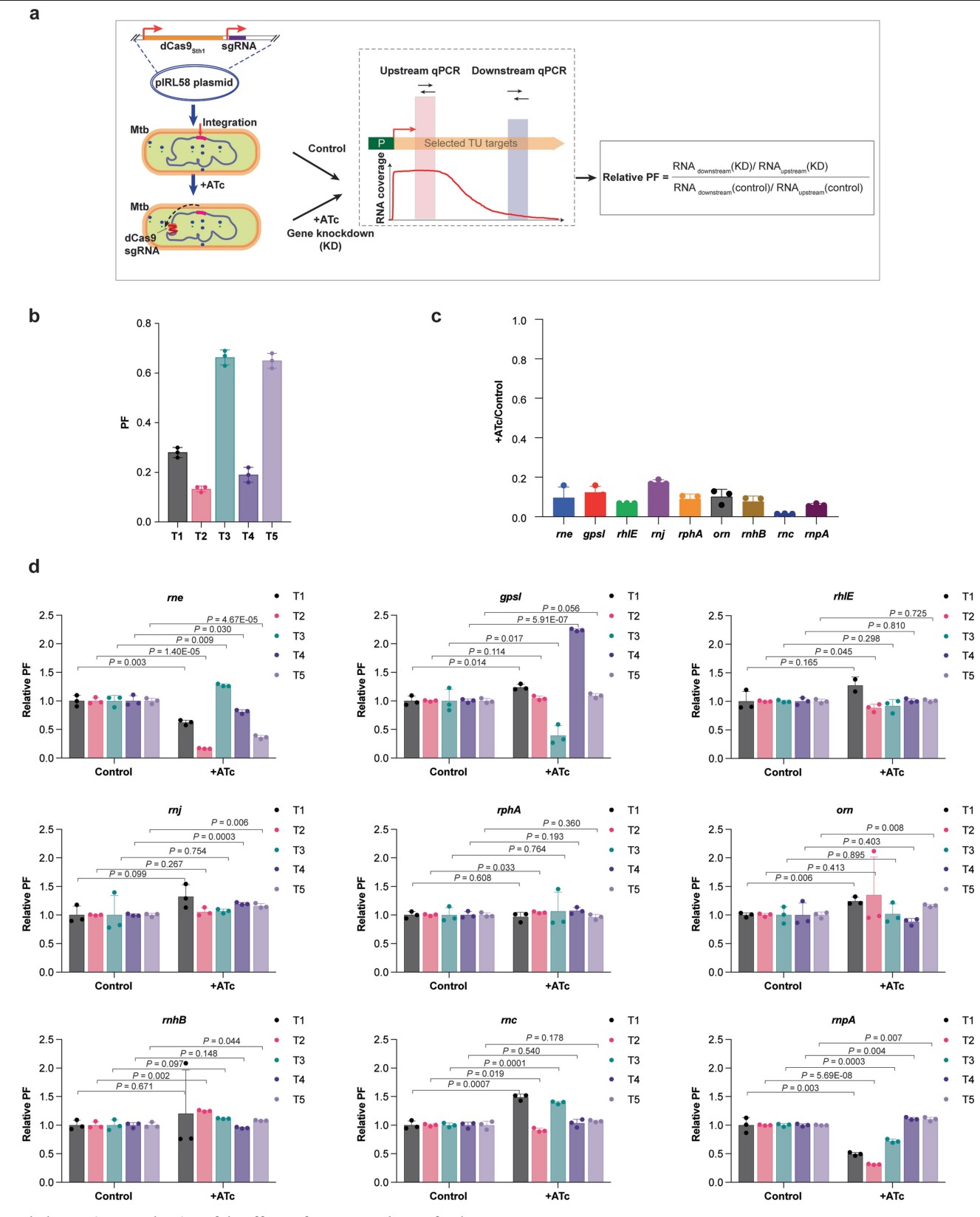

**Extended Data Fig. 5 | Evaluation of the effects of RNases on the PF of Mtb TUs. a**, Schematic of the experimental strategy to knockdown individual Mtb RNase genes using CRISPRi and measure the PF of target TUs using qPCR. PF was calculated relative to a control strain expressing scrambled sgRNA. **b**, PF values for five selected Mtb coding TUs in control cells. **c**, Validation of the knockdown efficiency for each Mtb RNase gene. **d**, Relative PF values for the five target TUs upon the individual knockdown of nine RNase or RNase-related genes, namely *rne*, *gpsI*, *rhlE*, *rnj*, *rphA*, *orn*, *rnhB*, *rnc*, and *rnpA*. Data are mean ± s.d. from three independent measurements. *P* values were determined using two-tailed Student's *t*-test.

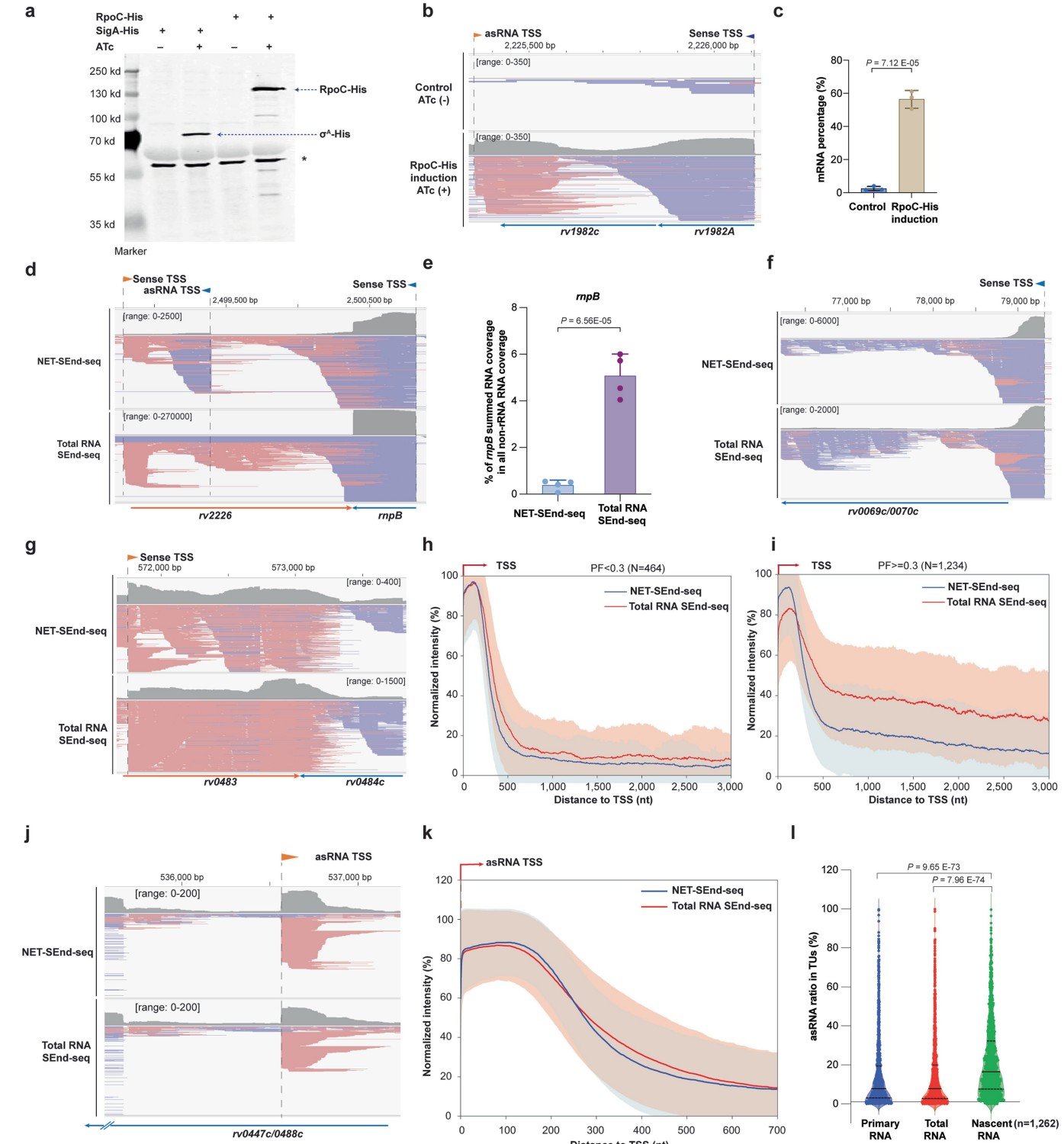

**Extended Data Fig. 6 |** See next page for caption.

**Extended Data Fig. 6 | Developing NET-SEnd-seq for nascent RNA profiling in Mtb. a**, Induced expression of His-tagged RpoC in Mtb cells verified by western blotting using anti-His antibody. His-tagged $\sigma^A$ was used as a positive control. Asterisk indicates a non-specific band. Molecular weight markers are shown on the left. This experiment was repeated twice with similar results. **b**, NET-SEnd-seq data track for an example Mtb TU with or without ATc-induced RpoC-His expression. **c**, Percentage of mRNAs in total cellular RNAs plus spike-in RNAs before and after ATc induction. The same amount of spike-in RNAs was added to each sample before library preparation. The significantly increased mRNA population upon RpoC-His expression indicates the specific enrichment of RNAP-bound nascent RNAs in the NET-SEnd-seq dataset. Data are mean ± s.d. from three independent measurements. $P$ value was determined using two-tailed Student's $t$-test. **d**, NET-SEnd-seq and total RNA SEnd-seq data tracks for the *rnpB* gene. **e**, Bar graphs comparing the percentage of *rnpB* mRNAs in total non-ribosomal RNAs for NET-SEnd-seq and total RNA SEnd-seq datasets. As expected for a highly abundant mRNA species, *rnpB* transcripts account for a significant fraction of total non-ribosomal RNAs. Its lower percentage in the NET-SEnd-seq dataset indicates the specific enrichment of nascent RNAs. Data are mean ± s.d. from four independent measurements. $P$ value was determined using two-tailed Student's $t$-test. **f-g**, SEnd-seq data tracks for an example low-PF TU (**f**) and an example high-PF TU (**g**) comparing the RNAs profiled by NET-SEnd-seq vs. total RNA SEnd-seq. **h-i**, Summed SEnd-seq intensities for TUs with a PF < 0.3 ($n = 464$) (**h**) or with a PF $\geq$ 0.3 ($n = 1,234$) (**i**) aligned at TSSs for nascent RNA (blue) and total RNA (red) isolated from log-phase Mtb cells. TUs that exhibited high expression within a 200-nt region downstream of the TSS in both datasets were selected for analysis. Colored lines represent median values and shaded regions represent s.d. **j**, NET-SEnd-seq and total RNA SEnd-seq data tracks for an example Mtb asRNA. **k**, Summed SEnd-seq intensities for asRNAs aligned at TSSs for nascent RNA (blue) and total RNA (red). asRNAs that exhibited high expression in both datasets were selected for analysis ($n = 1,852$). Colored lines represent median values and shaded regions represent s.d. **l**, Distribution of the percentage of asRNAs in each TU for primary RNA, total RNA, and nascent RNA isolated from log-phase Mtb cells. $P$ values were determined using two-tailed Student's $t$-test.

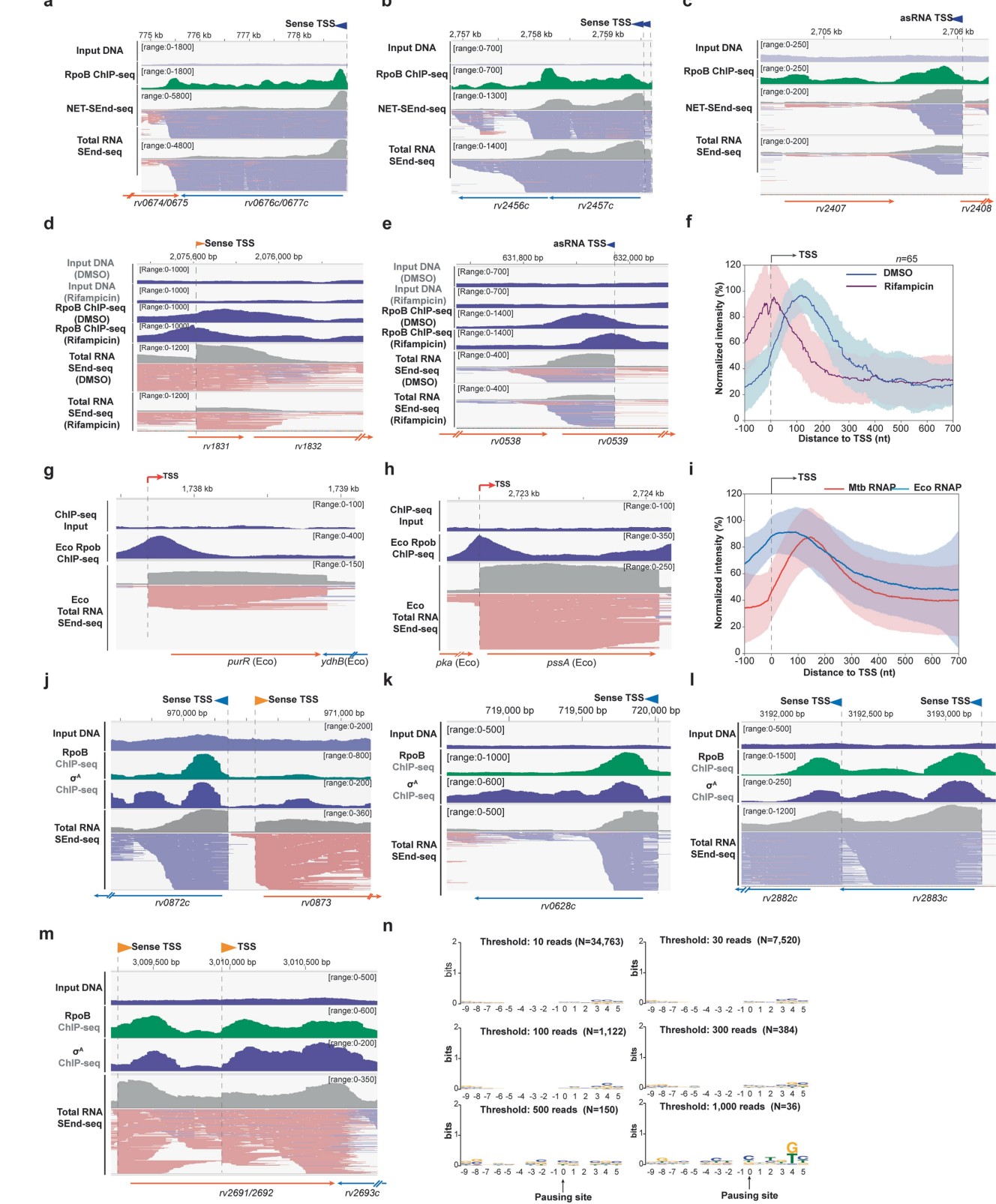

**Extended Data Fig. 7** | See next page for caption.

**Extended Data Fig. 7 | Additional Mtb RNAP and σ^A ChIP-seq results.**
**a-c**, Mtb RNAP ChIP-seq, NET-SEnd-seq, and total RNA SEnd-seq data tracks for three example Mtb genomic regions. Input DNA signals are shown on the top. **d-e**, Mtb RNAP ChIP-seq and total RNA SEnd-seq data tracks for two example Mtb genomic regions from cells treated with DMSO or rifampicin. **f**, Summed RNAP ChIP-seq intensities for transcribed regions aligned at TSSs from DMSO-treated or rifampicin-treated Mtb cells. TSSs without another strong TSS within 700 nt downstream and associated with strong RNAP occupancy in the rifampicin-treated condition were selected for analysis ($n = 65$). Colored lines represent median values and shaded regions represent s.d. **g-h**, Eco RNAP ChIP-seq and total RNA SEnd-seq data tracks for two example Eco genomic regions. **i**, Summed RNAP ChIP-seq intensities for TUs from log-phase Mtb and Eco cells. TSSs without another strong TSS within 700 nt downstream were selected for analysis ($n = 1,432$ for Mtb; $n = 932$ for Eco). Colored lines represent median values and shaded regions represent s.d. **j-m**, Mtb RNAP ChIP-seq and σ^A ChIP-seq data tracks for four example Mtb genomic regions and the corresponding total RNA SEnd-seq data tracks. Input DNA signals are shown on the top. Additional σ^A peaks that did not have a corresponding RNAP peak were sometimes observed, presumably due to RNAP-independent σ^A-–DNA interaction[51]. **n**, Motif analysis for DNA sequences surrounding the Mtb RNAP pause sites defined by the RNA 3′ ends in the NET-SEnd-seq data. Each panel uses a different read threshold. Positions with a higher read indicate stronger pause sites.

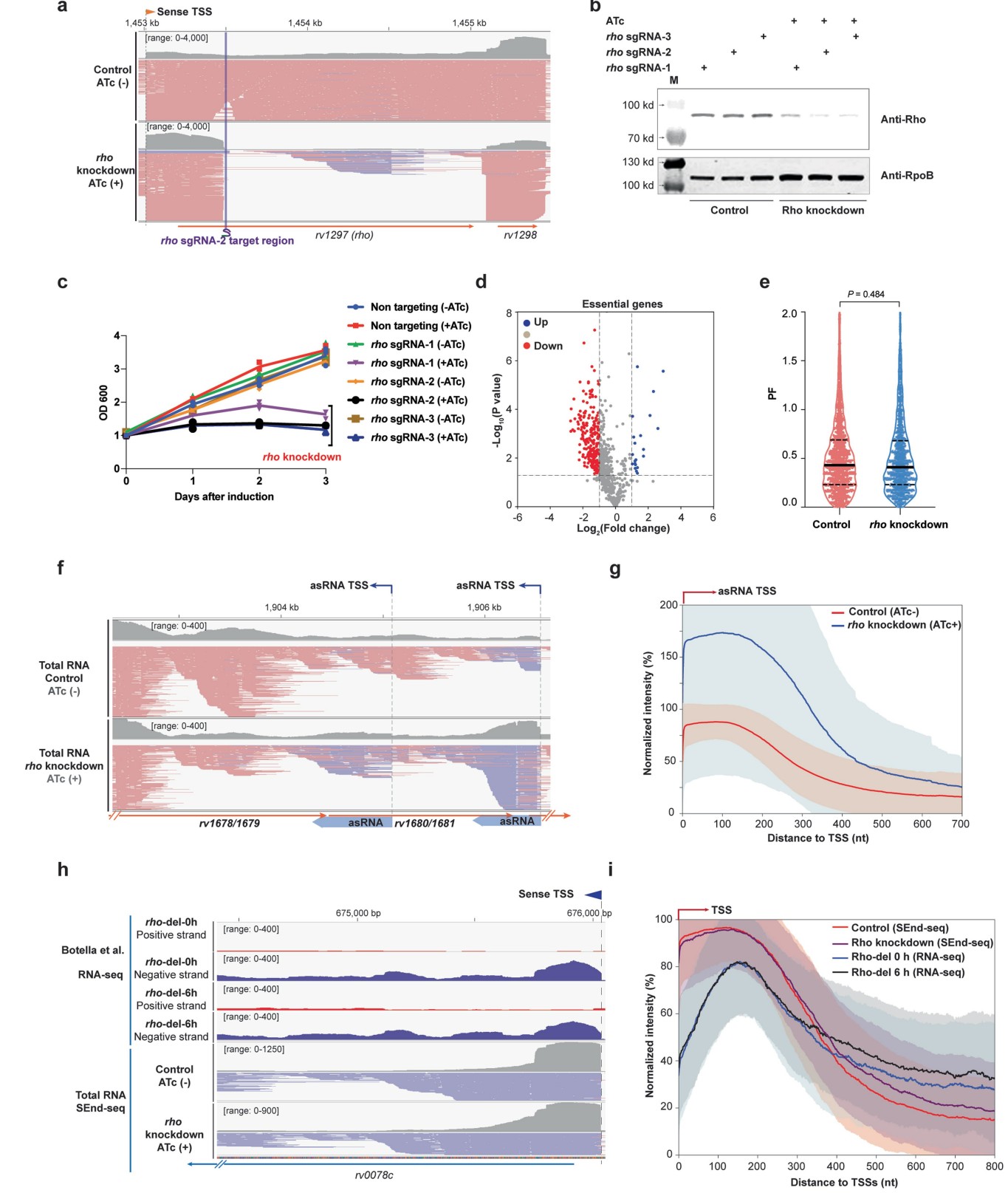

**Extended Data Fig. 8** | See next page for caption.

**Extended Data Fig. 8 | Effects of *rho* knockdown on the Mtb transcriptome.**
**a**, SEnd-seq data track around the Mtb *rho* gene (*Rv1297*) from cells with or
without ATc-induced sgRNA expression. The sgRNA targeting site is indicated
by the purple line. **b**, Depletion of Rho proteins upon sgRNA expression is
verified by western blotting. The RpoB signal serves as a loading control.
M indicates the molecular weight marker lane. **c**, Mtb growth curves with or
without ATc-induced *rho* knockdown. **d**, Volcano plot showing changes in
the expression level of essential Mtb genes (*n* = 696) upon *rho* knockdown
measured by SEnd-seq. *P* values were determined using two-tailed Student's
*t*-test. **e**, Violin plot showing the distribution of PFs for Mtb coding TUs with or
without *rho* knockdown. TUs that exhibited high expression within a 200-nt
window downstream of the TSS in both datasets were selected for analysis
(*n* = 1,679). The solid lines represent the median and the dashed lines represent
the quartiles. *P* value was determined using two-tailed Student's *t*-test. **f**, SEnd-
seq data track for an example Mtb genomic region showing an increase in
asRNA abundance upon *rho* knockdown. **g**, Summed SEnd-seq intensities for
Mtb asRNAs aligned at TSSs with or without *rho* knockdown. Highly expressed
asRNAs in the control condition were selected for analysis (*n* = 1,565). The
intensities for each position were normalized to the peak value in the control
group. **h**, Data track for an example Mtb TU showing the RNA-seq results before
and after Rho depletion from a previous study[18] and the corresponding total
RNA SEnd-seq results from the current study. **i**, Summed RNA-seq and SEnd-seq
intensities for Mtb TUs aligned at TSSs before and after Rho depletion. TUs
with a highly expressed upstream zone and a PF < 0.3 in the control condition
for wildtype cells were selected for analysis (*n* = 431). Colored lines represent
median values and shaded regions represent s.d.

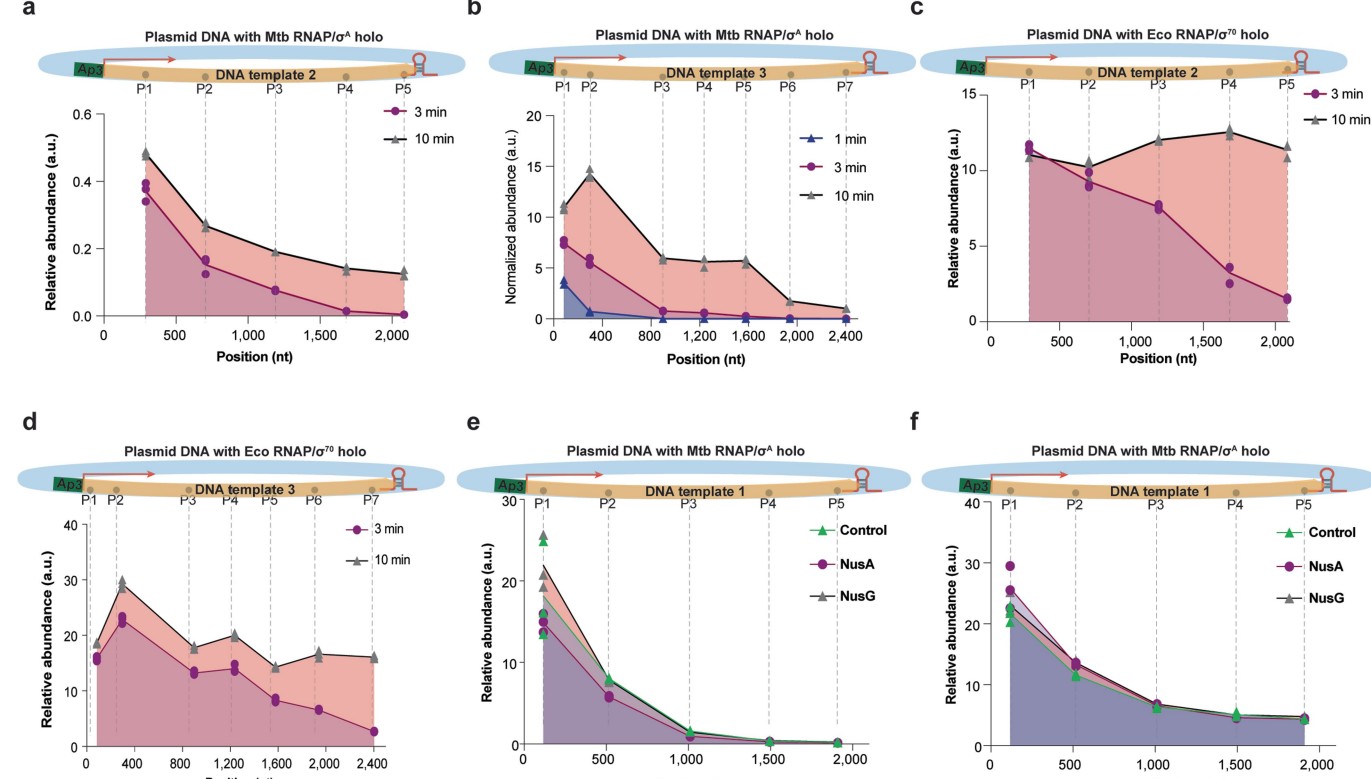

**Extended Data Fig. 9 | Additional in vitro transcription results. a-b**, Length profile of RNA products measured by qPCR using Mtb RNAP–σ^A holoenzyme and two plasmid DNA templates with different transcribed regions (template 2 in **a** and template 3 in **b**) than that used in Fig. 4a (template 1). **c-d**, Length profile of RNA products using Eco RNAP–σ^70 holoenzyme and plasmid DNA templates 2 (**c**) and 3 (**d**). **e**, Length profile of RNA products using plasmid DNA template 1 and Mtb RNAP–σ^A holoenzyme alone (control) or in the presence of Mtb NusA or NusG measured at the 3-min time point. **f**, Same as **e** except that the RNA abundances were measured at the 10-min time point. Data are from three independent measurements.

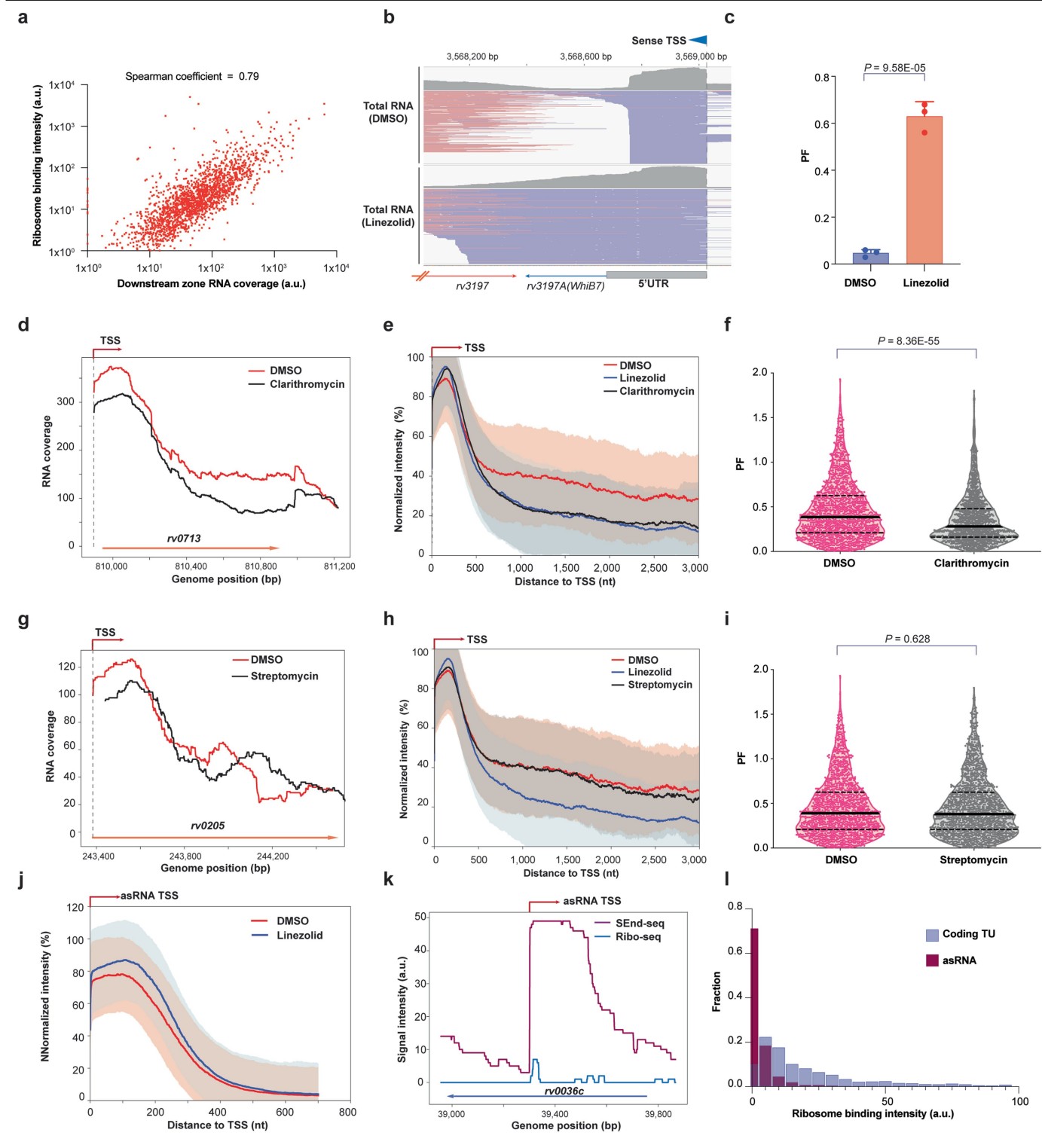

**Extended Data Fig. 10** | See next page for caption.

**Extended Data Fig. 10 | Effects of translation inhibitors on transcription elongation in Mtb. a**, Scatter plot showing the correlation between the ribosome binding intensity for Mtb coding TUs (measured by ribo-seq and normalized by the TU length) and the corresponding RNA coverage in the TU's downstream zone (measured by SEnd-seq and normalized by the length of the downstream zone from 500 nt downstream of TSS to the end of TU). TUs that exhibited high expression within a 200-nt window downstream of the TSS in the SEnd-seq dataset were selected for analysis ($n = 1,855$). Spearman correlation coefficient is shown. **b**, Total RNA SEnd-seq data track for the *whiB7* gene from Mtb cells treated with DMSO or linezolid. **c**, Bar graphs comparing the PF for *whiB7* transcription from Mtb cells treated with DMSO or linezolid. Data are mean ± s.d. from three independent measurements. *P* value was determined using two-tailed Student's *t*-test. **d**, SEnd-seq signals for an example TU from Mtb cells treated with clarithromycin (black) and control cells treated with DMSO (red). **e**, Summed SEnd-seq intensities aligned at TSSs for DMSO-treated (red), linezolid-treated (blue), and clarithromycin-treated (black) Mtb cells. The RNA intensities for each position were normalized to the peak value in the corresponding condition. Highly expressed TUs in the DMSO-treated condition with a PF > 0.3 were selected for analysis ($n = 1,144$). Colored lines represent median values and shaded regions represent s.d. **f**, Distribution of PFs for TUs from DMSO-treated or clarithromycin-treated Mtb cells. TUs that exhibited high expression within a 200-nt window downstream of the TSS in both datasets were selected for analysis ($n = 1,543$). The solid lines represent the median and the dashed lines represent the quartiles. *P* value was determined using two-tailed Student's *t*-test. **g**, SEnd-seq signals for an example TU from Mtb cells treated with streptomycin (black) and control cells treated with DMSO (red). **h**, Summed SEnd-seq intensities aligned at TSSs for DMSO-treated (red), linezolid-treated (blue), and streptomycin-treated (black) Mtb cells. The RNA intensities for each position were normalized to the peak value in the corresponding condition. Highly expressed TUs in the DMSO-treated condition with a PF > 0.3 were selected for analysis ($n = 1,144$). Colored lines represent median values and shaded regions represent s.d. **i**, Distribution of PFs for TUs from DMSO-treated or streptomycin-treated Mtb cells. TUs that exhibited high expression within a 200-nt window downstream of the TSS in both datasets were selected for analysis ($n = 1,535$). The solid lines represent the median and the dashed lines represent the quartiles. *P* value was determined using two-tailed Student's *t*-test. **j**, Summed SEnd-seq intensities for antisense RNAs from DMSO-treated (red) and linezolid-treated (blue) Mtb cells aligned at TSSs. Highly expressed asRNAs in the DMSO-treated condition were selected for analysis ($n = 1,370$). Colored lines represent median values and shaded regions represent s.d. **k**, SEnd-seq and ribo-seq signals for an example Mtb asRNA. **l**, Distribution of ribosome binding intensities for highly expressed Mtb coding TUs ($n = 1,685$) and asRNAs ($n = 2,162$) (normalized by their lengths). Only units with a normalized ribosome binding intensity of less than 100 are shown here.

# Reporting Summary

## Statistics

For all statistical analyses, confirm that the following items are present in the figure legend, table legend, main text, or Methods section.

| n/a | Confirmed | |
|---|---|---|
| ☐ | ☒ | The exact sample size (*n*) for each experimental group/condition, given as a discrete number and unit of measurement |
| ☐ | ☒ | A statement on whether measurements were taken from distinct samples or whether the same sample was measured repeatedly |
| ☐ | ☒ | The statistical test(s) used AND whether they are one- or two-sided *Only common tests should be described solely by name; describe more complex techniques in the Methods section.* |
| ☒ | ☐ | A description of all covariates tested |
| ☒ | ☐ | A description of any assumptions or corrections, such as tests of normality and adjustment for multiple comparisons |
| ☐ | ☒ | A full description of the statistical parameters including central tendency (e.g. means) or other basic estimates (e.g. regression coefficient) AND variation (e.g. standard deviation) or associated estimates of uncertainty (e.g. confidence intervals) |
| ☐ | ☒ | For null hypothesis testing, the test statistic (e.g. *F*, *t*, *r*) with confidence intervals, effect sizes, degrees of freedom and *P* value noted *Give P values as exact values whenever suitable.* |
| ☒ | ☐ | For Bayesian analysis, information on the choice of priors and Markov chain Monte Carlo settings |
| ☒ | ☐ | For hierarchical and complex designs, identification of the appropriate level for tests and full reporting of outcomes |
| ☐ | ☒ | Estimates of effect sizes (e.g. Cohen's *d*, Pearson's *r*), indicating how they were calculated |

*Our web collection on statistics for biologists contains articles on many of the points above.*

## Software and code

Policy information about availability of computer code

| | |
|---|---|
| Data collection | The sequencing data were collected on Illumina platforms, including NextSeq 500, NovaSeq 6000, at The Rockefeller University Genomics Resource Center. |
| Data analysis | After quality filtering and Illumina sequencing adaptor trimming with FASTX-Toolkit (v0.0.13) , the raw paired-end reads were merged to single-end reads by using FLASh software (v1.2.11, T. Magoc et al., 2011). The correlated 5'-end and 3'-end sequences were extracted by the custom script ('fasta_to_paired.sh') utilizing the SeqKit (v2.4.0) and Cutadapt (v4.1) packages. The inferred full-length reads were generated by Bedtools (v2.31.0, Quinlan et al., 2010) and Samtools (v1.17, Danecek et al., 2021) after mapping to the reference genome (NC000913.3 for Eco, NC008596.1 for Msm, and NC018143.2 for Mtb) via Bowtie2 (v2.5.1, Langmead et al., 2012). The full-length reads with an insert length greater than 10,000 nt were discarded. The mapping results were visualized using the  IGV genome viewer (v2.4.10, Robinson et al., 2011). SEnd-seq coverage was calculated per nucleotide position in the genome, and further analyses were performed using Perl (v5.34.1) or Python (3.11.3) with custom scripts available on Github (https://github.com/LiuLab-codes/Mtb_transcriptome_profiling). Data analysis and visualization scripts utilized Python packages including Matplotlib (v3.7.1), Numpy (v1.24.3), Scipy (v1.10.1), bioinfokit (v0.3), and pyCircos (v0.3.0). Motif analysis was conducted using the MEME suite (v5.5.2, Bailey et al., 2015). The Gene Ontology analysis was performed on DAVID website (v2023q2, Sherman et al., 2022). |

For manuscripts utilizing custom algorithms or software that are central to the research but not yet described in published literature, software must be made available to editors and reviewers. We strongly encourage code deposition in a community repository (e.g. GitHub). See the Nature Portfolio guidelines for submitting code & software for further information.

## Data

Policy information about availability of data

All manuscripts must include a data availability statement. This statement should provide the following information, where applicable:

- Accession codes, unique identifiers, or web links for publicly available datasets
- A description of any restrictions on data availability
- For clinical datasets or third party data, please ensure that the statement adheres to our policy

> SEnd-seq and ChIP-seq datasets from this study are deposited in Gene Expression Omnibus (GEO) with the accession number GSE211992 (BioProject PRJNA873109).

## Human research participants

Policy information about studies involving human research participants and Sex and Gender in Research.

| Reporting on sex and gender | N/A |
| --- | --- |
| Population characteristics | N/A |
| Recruitment | N/A |
| Ethics oversight | N/A |

Note that full information on the approval of the study protocol must also be provided in the manuscript.

# Field-specific reporting

Please select the one below that is the best fit for your research. If you are not sure, read the appropriate sections before making your selection.

☒ Life sciences        ☐ Behavioural & social sciences        ☐ Ecological, evolutionary & environmental sciences

For a reference copy of the document with all sections, see nature.com/documents/nr-reporting-summary-flat.pdf

# Life sciences study design

All studies must disclose on these points even when the disclosure is negative.

| Sample size | Sequencing experiments were conducted at least twice for each condition, and all other experiments were repeated at least three times, unless specified otherwise. For RNA samples, we collected from 4 mL of log-phase  cells, and for ChIP-seq, samples were taken from 50 mL of log-phase cells. The SEnd-seq sample was sequenced to a depth more than 8 million reads, and the ChIP-seq sample to a depth of 3-10 million reads. These depths are sufficient for a comprehensive characterization of the transcriptome and ChIP signals in each sample, aligning with standard practices in the field. |
| --- | --- |
| Data exclusions | Low-quality reads and any paired-end sequences with an alignment insert length exceeding 10,000 nucleotides were excluded from subsequent data analysis. Other specific criteria are described in figure legends. |
| Replication | Sequencing experiments were conducted at least twice for each condition, and all other experiments were repeated at least three times, unless otherwise specified. 3 technical replicates were performed for qPCR measurements. |
| Randomization | This study didn't include experiments that required randomization, as there was no allocation of samples into groups. |
| Blinding | This study didn't include experiments that required blinding, as all experiments were quantitative and there was no allocation of samples into groups. |

# Reporting for specific materials, systems and methods

We require information from authors about some types of materials, experimental systems and methods used in many studies. Here, indicate whether each material, system or method listed is relevant to your study. If you are not sure if a list item applies to your research, read the appropriate section before selecting a response.

## Materials & experimental systems

| n/a | Involved in the study |
|-----|----------------------|
| ☐ | ☒ Antibodies |
| ☒ | ☐ Eukaryotic cell lines |
| ☒ | ☐ Palaeontology and archaeology |
| ☒ | ☐ Animals and other organisms |
| ☒ | ☐ Clinical data |
| ☒ | ☐ Dual use research of concern |

## Methods

| n/a | Involved in the study |
|-----|----------------------|
| ☐ | ☒ ChIP-seq |
| ☒ | ☐ Flow cytometry |
| ☒ | ☐ MRI-based neuroimaging |

# Antibodies

| | |
|---|---|
| Antibodies used | Antibodies against Mtb Rho (a gift from D. Schnappinger, Weill Cornell Medicine), against Eco RpoB ( BioLegend, 663903, developed with a peptide fragment common to bacterial RpoB), against Eco sigma 70 (BioLegend, 663208, developed with a peptide fragment of the common bacterial housekeeping sigma factor) and against His-Tag (Santa cruze, sc-8036) were used. |
| Validation | The antibody against Mtb Rho was previously validated by western blot (PMID:28348398). The antibody against Eco RpoB was previously used in Mtb RpoB ChIP-seq experiments (PMID: 23222129) and validated by western blot (PMID: 30242166). The antibody against Eco Sigma70 was previously used in Msm SigA ChIP-seq experiments (PMID: 25089258). The antibody against His-Tag was previously used in western blot experiments (PMID: 37154023). |

# ChIP-seq

## Data deposition

☒ Confirm that both raw and final processed data have been deposited in a public database such as GEO.

☒ Confirm that you have deposited or provided access to graph files (e.g. BED files) for the called peaks.

| | |
|---|---|
| Data access links<br>*May remain private before publication.* | SEnd-seq, NET-SEnd-seq, and ChIP-seq datasets from this study are deposited in Gene Expression Omnibus (GEO) with the accession number GSE211992 (BioProject PRJNA873109). |
| Files in database submission | ChIP_seq_WT_input_rep1_R1_001.fastq.gz ChIP_seq_WT_input_rep1_R2_001.fastq.gz<br>ChIP_seq_WT_input_rep2_R1_001.fastq.gz ChIP_seq_WT_input_rep2_R2_001.fastq.gz<br>ChIP_seq_WT_Ab_RpoB_rep1_R1_001.fastq.gz ChIP_seq_WT_Ab_RpoB_rep1_R2_001.fastq.gz<br>ChIP_seq_WT_Ab_RpoB_rep2_R1_001.fastq.gz ChIP_seq_WT_Ab_RpoB_rep2_R2_001.fastq.gz<br>ChIP_seq_WT_DMSO_input_rep1_R1_001.fastq.gz ChIP_seq_WT_DMSO_input_rep1_R2_001.fastq.gz<br>ChIP_seq_WT_RIF_input_rep1_R1_001.fastq.gz ChIP_seq_WT_RIF_input_rep1_R2_001.fastq.gz<br>ChIP_seq_WT_DMSO_Ab_RpoB_rep1_R1_001.fastq.gz ChIP_seq_WT_DMSO_Ab_RpoB_rep1_R2_001.fastq.gz<br>ChIP_seq_WT_DMSO_Ab_RpoB_rep2_R1_001.fastq.gz ChIP_seq_WT_DMSO_Ab_RpoB_rep2_R2_001.fastq.gz<br>ChIP_seq_WT_RIF_Ab_RpoB_rep1_R1_001.fastq.gz ChIP_seq_WT_RIF_Ab_RpoB_rep1_R2_001.fastq.gz<br>ChIP_seq_WT_RIF_Ab_RpoB_rep2_R1_001.fastq.gz ChIP_seq_WT_RIF_Ab_RpoB_rep2_R2_001.fastq.gz<br>ChIP_seq_WT_Ab_SigA_rep1_R1_001.fastq.gz ChIP_seq_WT_Ab_SigA_rep1_R2_001.fastq.gz<br>ChIP_seq_WT_Ab_SigA_rep2_R1_001.fastq.gz ChIP_seq_WT_Ab_SigA_rep2_R2_001.fastq.gz<br>ChIP_seq_WT_Ab_SigA_rep3_R1_001.fastq.gz ChIP_seq_WT_Ab_SigA_rep3_R2_001.fastq.gz |
| Genome browser session<br>(e.g. UCSC) | No longer applicable. |

## Methodology

| | |
|---|---|
| Replicates | At least two replicates were prepared for each sample. |
| Sequencing depth | WT_input_rep1, pair-end, read number: 1648782, read-length: 75 bp<br>WT_input_rep2, pair-end, read number: 1447153, read-length: 75 bp<br>ChIP_seq_WT_Ab_RpoB_rep1, pair-end, read number:10254546, read-length: 150 bp<br>ChIP_seq_WT_Ab_RpoB_rep2, pair-end, read number:10799500, read-length: 150 bp<br>ChIP_seq_WT_Ab_SigA_rep1, pair-end, read number: 4971901, read-length: 150 bp<br>ChIP_seq_WT_Ab_SigA_rep2, pair-end, read number: 5360395, read-length: 150 bp<br>ChIP_seq_WT_Ab_SigA_rep3, pair-end, read number: 4896344, read-length: 150 bp<br>WT_DMSO_input, pair-end, read number: 3740538, read-length: 75 bp<br>WT_RIF_input, pair-end, read number: 3445471, read-length: 75 bp<br>ChIP_seq_WT_DMSO_Ab_RpoB_rep1, pair-end, read number: 7397067, read-length: 75 bp<br>ChIP_seq_WT_DMSO_Ab_RpoB_rep2, pair-end, read number: 7931837, read-length: 75 bp<br>ChIP_seq_WT_RIF_Ab_RpoB_rep1, pair-end, read number:7954605, read-length: 75 bp<br>ChIP_seq_WT_RIF_Ab_RpoB_rep2, pair-end, read number:6939423, read-length: 75 bp |
| Antibodies | The antibody against Eco RpoB (BioLegend, 663903) was used in the Mtb RpoB ChIP-seq experiments. The antibody against Eco Sigma70 (BioLegend, 663208) was used in the Mtb SigA ChIP-seq experiments. |

| Peak calling parameters | We analyzed the signal coverage with custom scripts. |
| --- | --- |
| Data quality | Reads with high quality (quality score >=30)  were retained using FASTX-Toolkit (v0.0.13) and only uniquely mapped reads were used for downstream analysis. |
| Software | Samtools (version 1.18), Bowtie2 (version 2.5.1), Bedtools (version 2.31.0), Python (version 3.11.3),  Matplotlib (v3.7.1), Numpy (v1.24.3), and custom scripts were used. |

