## [Peer Review File · Nature]

Manuscript Title: Incomplete transcripts dominate the Mycobacterium tuberculosis transcriptome

Editorial Notes:

Reviewer Comments & Author Rebuttals

Reviewer Reports on the Initial Version:

Referees' comments:

Referee #1 (Remarks to the Author):

In this manuscript, Ju et al. leverage their SEnd-seq sequencing protocol to study the transcriptome of the human pathogen *M. tuberculosis*. SEnd-seq unveils, on a genome-wide scale and at a single nucleotide resolution, the 5' and 3' ends of the same RNA molecule. The prowess of the method emanates from the stratification of RNA molecules to primary transcripts, processed species, and degradation products. As such, the authors are able to accurately mark sites of transcription initiation, as well as termination. Previously the authors used SEnd-seq to discover bi-directional transcription termination sites (TTS) in the *E. coli* genome. Herein, they apply SEnd-seq to study the transcriptome of *M. tuberculosis*, and the findings highlight a new facet of this dreadful pathogen. First, the authors report a high number of transcriptions start sites (TSS); Although the genome size of *M. tuberculosis* is roughly equivalent to the *E. coli* genome, it is furnished with nearly 3-fold more TSS. In contrast, there are far fewer TTS. Second, on top of the unusually high TTS/TSS ratio, the authors discover that a sizeable number of transcripts are incomplete in *M. tuberculosis*. The authors suggest that low processivity of the transcription machinery accounts for the high abundance of such short transcripts. Using an inducible lacZ model and RT-qPCR assays, the authors reproduce the same trend of slow accumulation of full-length transcript against a backdrop of rapid increase in the level of short transcripts. Moreover, in a purified transcription system with the core-enzyme and vegetative sigma factor, sigma A, of *M. tuberculosis*, the authors demonstrate low processivity of its transcription machinery. Additional ChIP-seq and ribo-seq experiments support a key role for transcription-translation coupling in the synthesis of full-length transcripts.

The SEnd-seq method is elegant and the findings are very interesting. Yet, it remains to be seen what are the physiological consequences of the low processivity of the transcription machinery in *M. tuberculosis*. There are a few major comments I would like the authors to consider.

Major comments:

1. Transcription termination sites. The authors report a surprisingly low number of TTS across the *M. tuberculosis* genome. Is it possible that SEnd-seq detects with low sensitivity terminators of low-expression genes? I am asking because the high prevalence of shorter transcripts might hinder the lower

abundance of full transcripts. This might be especially relevant with low-expression genes that have fewer reads to begin with. The authors might want to stratify the terminators they detect based on gene-expression level from total RNA-seq. In case terminators are detected mostly for highly-expressed genes, the authors might want to discuss this limitation of SEnd-seq.

2. Shorter transcripts. The authors do not elaborate enough on the distinction between nascent to non-nascent transcripts (with the exception of the sentence: “Therefore, the short RNAs in the Mtb transcriptome are more likely generated during nascent RNA synthesis than during post transcriptional RNA processing”). The authors should provide a clear answer to this important question: are the short transcripts captured from paused elongating transcription complexes? Or are these post-termination RNA species which were released from RNAP? The authors might want to use nascent RNA after pull-down of RNAP complexes as input for SEnd-seq to address this question.

3. Proteomic analysis of Linezolid treated cells. Contingent on the answer to my comment above, truncated proteins are expected to accumulate in Linezolid treated cells, when transcription-translation coupling is disturbed. Can the authors detect an enrichment of truncated proteins under any condition in *M. tuberculosis*, and not for example in *M. smegmatis* where PF values are higher at baseline?

4. The role of sigmaA in the low processivity of the transcription machinery of *M. tuberculosis*. In Figure 3 and Extended Data Figure 10, the authors present length analysis of RNA products from in vitro transcription reactions. Whenever sigma A is bound to the core enzyme, it has an adverse effect on transcription processivity. This result is especially striking with a scaffold, with a pre-formed bubble, that obviates the need for sigma. Still, it is not clear whether the in vitro results emulate the behavior of sigma in living cells. Do the authors record a different gene-PF values depending on which sigma factor is transcribing a given gene in *M. tuberculosis*? How does the closely-related *M. smegmatis* sigma factor affect transcription processivity in the same type of assay?

5. Recent publications suggest that incomplete transcripts dominate the transcriptome of other bacteria species, including that of *E. coli*. The authors should cite the work of Herzel et al. (PMID: 35524564).

Minor

“...and Eco RNAP can retain sigma70 throughout the transcription cycle in some contexts” the authors may cite an earlier paper here as well (PMID: 11525730).

Referee #2 (Remarks to the Author):

The authors use SEnd-seq to map full-length transcripts in *Mycobacterium tuberculosis* (Mtb). They

show that for most genes, the transcripts produced are frequently shorter than the gene, suggesting incomplete transcription. This is in contrast to the situation in *Escherichia coli*, where most transcripts appear to be full-length. The authors argue that Mtb RNA polymerase has lower processivity than RNAP polymerase from *E. coli*, and that this difference explains why Mtb transcripts are often shorter than the associated genes. Lastly, the authors show that impaired translation leads to production of even shorter transcripts for protein-coding genes, suggesting that transcription processivity is enhanced by coupled translation.

The observation that Mtb transcripts are often incomplete is interesting because of the potential physiological implications for translation. Unfortunately, the authors do not address the impact of pervasive premature transcription on Mtb physiology. One important area of investigation would be the impact on tmRNA usage. With so many prematurely terminated transcripts for mRNAs, you would expect an outsized role for tmRNA in Mtb since tmRNA is involved in dealing with non-stop translation (i.e., mRNAs with a start codon but no stop codon, where otherwise ribosomes would become trapped at the 3' end). It would also be interesting to look at the impact of premature transcription termination on ORF and/or operon size, and gene position within operons.

In addition, there is insufficient evidence to conclude that RNA polymerase processivity is the cause of frequent incomplete transcripts. Additionally, the authors have not ruled out a role for Rho-dependent polarity in the reduced processivity of transcription under conditions of translation inhibition.

Assuming that the pervasiveness of incomplete transcripts is indeed due to the compromised RNA polymerase processivity, the impact of the paper could be increased by determining the mechanism that explains why Mtb RNA polymerase is less processive than its *E. coli* counterpart. The authors present some evidence that implicates the Sigma factor. This is an intriguing possibility, but there are no data presented that can explain how the Sigma factor impacts transcription elongation, and there is extensive evidence that most Sigma factor is rapidly released from elongating RNA polymerase in vivo.

Major comments:

1. The authors use CHIP-seq of RNA polymerase to make the argument that Mtb RNA polymerase is less processive than RNA polymerase from *E. coli*. However, they do not present a comparison of Mtb RNA polymerase CHIP-seq data to *E. coli* RNA polymerase CHIP-seq data. And in fact, a previous study (PMID 19150431; not cited in the current study) showed a very similar profile of RNA polymerase CHIP-seq in *E. coli*, suggesting that processivity of *E. coli* and Mtb RNA polymerases are similar in vivo. In vitro data support the authors' claim; however, there are many caveats associated with these data that are not discussed, including the lack of elongation factors (e.g., NusA, NusG), the potential that differences in reaction conditions could impact processivity, and the fact that only one template is used to compare *E. coli* and Mtb RNA polymerases.
2. Figure 4 looks at the effect of translation inhibition on transcription processivity. The authors conclude that inhibiting translation causes a reduction in transcription processivity on mRNAs, independent of Rho. However, they do not test whether depleting Rho reverses the effect of translation

inhibition on transcription processivity. Hence, there are insufficient data to conclude that the effect of translation inhibition on transcription processivity is independent of Rho.

Additional comments:

1. The authors describe two different SEnd-seq methods in the methods section. One method enriches for primary transcripts while the other does not. The authors do not indicate which method was used for the data presented in the paper. This is particularly important since it impacts how some of the data are interpreted. For example, Figure 1b shows many short transcripts associated with the Rv0517 gene that the authors attribute to premature transcription termination. If these data are from primary transcripts only, it is possible that there are other (processed) transcripts whose 5' ends align with the 3' ends of the short transcripts that initiate at the TSS. Such a result would strongly suggest a role for RNase processing. Assuming the authors have data for both methods, they should show the data side by side.
2. The authors show that inducing expression of lacZ transcription in Mtb leads to accumulation of RNA primarily at the 5' end of the gene. They argue that these data support a model in which short transcripts are generated due to premature transcription termination rather than RNA degradation, but the logic for this argument is not clearly explained.
3. SEnd-seq is a good approach to tackle the questions posed here, but RNA-seq would also suffice to determine whether there is widespread premature transcription termination. The authors should compare their SEnd-seq data to some of the many published RNA-seq datasets for Mtb. If the authors' model is correct, you would expect to see a similar drop-off in RNA-seq reads through transcribed regions as seen with SEnd-seq.
4. ChIP-seq of RNA polymerase is a reasonable approach for mapping the distribution of RNA polymerase across a transcribed region. However, it cannot distinguish between signal from initiating RNA polymerase and signal from elongating RNA polymerase. NET-seq would be a better approach since it looks only at elongating RNAP, and it provides single nucleotide resolution.
5. The authors compare their Mtb SEnd-seq data to E. coli data, but they do not mention the source of the E. coli data. Are the E. coli data the ones previously published in reference 3?
6. The authors should compare the TSSs they identify to those identified by two earlier studies (reference 8 and PMID 26536359). They could also look to see if the putative TSSs have the expected promoter sequences upstream. This would increase confidence in the TSS assignments.
7. The authors assess the position of Mtb TSSs relative to genes and then compare these data to data for other bacterial species. They conclude that Mtb has more antisense RNAs than other species. I would be hesitant draw a strong conclusion given that small technical differences can have an outsized impact on TSS identification (see PMID 25266388).
8. The authors should compare the list of 747 putative leaderless ORFs they identify based on SEnd-seq data to leaderless ORFs identified in earlier studies ((reference 8 and PMID 26536359). The example they show in Extended Data Figure 2f has been described previously (reference 8) as a leaderless ORF.
9. The authors should mention that a previous study (reference 8) also detected extensive antisense transcription in Mtb.
10. The authors should compare their SEnd-seq data from cells with depleted Rho to RNA-seq data in

reference 13. There appear to be some substantial differences in the datasets between the two studies; data in reference 13 are consistent with transcripts being extended in cells with depleted Rho.

11. In the Discussion, the authors hypothesize that translation can "push" transcription, increasing processivity. They suggest that this phenomenon does not occur in *E. coli*. However, reference 14 provides evidence for exactly this mechanism in *E. coli*.

Author Rebuttals to Initial Comments:

Referees' comments:

Referee #1 (Remarks to the Author):

In this manuscript, Ju et al. leverage their SEnd-seq sequencing protocol to study the transcriptome of the human pathogen *M. tuberculosis*. SEnd-seq unveils, on a genome-wide scale and at a single nucleotide resolution, the 5' and 3' ends of the same RNA molecule. The prowess of the method emanates from the stratification of RNA molecules to primary transcripts, processed species, and degradation products. As such, the authors are able to accurately mark sites of transcription initiation, as well as termination. Previously the authors used SEnd-seq to discover bi-directional transcription termination sites (TTS) in the *E. coli* genome. Herein, they apply SEnd-seq to study the transcriptome of *M. tuberculosis*, and the findings highlight a new facet of this dreadful pathogen. First, the authors report a high number of transcriptions start sites (TSS); Although the genome size of *M. tuberculosis* is roughly equivalent to the *E. coli* genome, it is furnished with nearly 3-fold more TSS. In contrast, there are far fewer TTS. Second, on top of the unusually high TTS/TSS ratio, the authors discover that a sizeable number of transcripts are incomplete in *M. tuberculosis*. The authors suggest that low processivity of the transcription machinery accounts for the high abundance of such short transcripts. Using an inducible *lacZ* model and RT-qPCR assays, the authors reproduce the same trend of slow accumulation of full-length transcript against a backdrop of rapid increase in the level of short transcripts. Moreover, in a purified transcription system with the core-enzyme and vegetative sigma factor, sigma A, of *M. tuberculosis*, the authors demonstrate low processivity of its transcription machinery. Additional ChIP-seq and ribo-seq experiments support a key role for transcription-translation coupling in the synthesis of full-length transcripts.

The SEnd-seq method is elegant and the findings are very interesting. Yet, it remains to be seen what are the physiological consequences of the low processivity of the transcription machinery in *M. tuberculosis*. There are a few major comments I would like the authors to consider.

Response: We are grateful to this reviewer for their excellent summary and favorable opinion on our work. We have further improved the manuscript by addressing the major comments with new experiments and analyses as detailed below. With the new data, our study describes both the phenomenon and mechanism of prevalent RNAP pausing that dominates the Mtb transcriptome, resembling the well-known promoter-proximal pausing of eukaryotic RNA polymerase II. As such, this work establishes a new paradigm for understanding bacterial transcription that is remarkably distinct from the existing one largely drawn from *E. coli* studies. We have added an expanded discussion on the implications of our findings for Mtb gene regulation and TB therapy in the revised manuscript. I hope the reviewer would agree with us that, as we have witnessed in the eukaryotic transcription field, it will likely take years or even decades of research to fully address the physiological consequences of promoter-proximal pausing in bacteria, which goes well beyond the scope of this paper.

Major comments:

1. Transcription termination sites. The authors report a surprisingly low number of TTS across the *M. tuberculosis* genome. Is it possible that SEnd-seq detects with low sensitivity terminators of low-expression genes? I am asking because the high prevalence of shorter transcripts might hinder the lower abundance of full transcripts. This might be especially relevant with low-expression genes that have fewer reads to begin with. The authors might want to stratify the terminators they detect based on gene-expression level from total RNA-seq. In case terminators are detected mostly for highly-expressed genes, the authors might want to discuss this limitation of SEnd-seq.

Response: We thank the reviewer for bringing up this important point. The criteria we used to define a TTS is as follows: “Potential TTSs were identified from the total RNA SEnd-seq data at genomic positions with more than 10 reads ending at that position (outside rRNA genes) and with a reduction of more than 40% in read coverage from its upstream to its downstream (e.g., 100 reads at position -1 and 50 reads at position 0).” Indeed, we cannot detect TTS for very lowly expressed genes (regardless of the fraction of full-length transcripts for that gene) due to this cutoff. We binned the Mtb TTSs detected by SEnd-seq based on the corresponding gene expression level (now shown as new Extended Data Fig. 3d) and did not find a bias towards highly expressed genes. We note that the paucity of TTSs in the Mtb genome has also been reported by previous studies, now cited in the revised manuscript (Line 68). We now mention the caveat of our TTS detection criteria in the caption of Extended Data Fig. 3.

2. Shorter transcripts. The authors do not elaborate enough on the distinction between nascent to non-nascent transcripts (with the exception of the sentence: “Therefore, the short RNAs in the Mtb transcriptome are more likely generated during nascent RNA synthesis than during post transcriptional RNA processing”). The authors should provide a clear answer to this important question: are the short transcripts captured from paused elongating transcription complexes? Or are these post-termination RNA species which were released from RNAP? The authors might want to use nascent RNA after pull-down of RNAP complexes as input for SEnd-seq to address this question.

Response: We appreciate this comment and fully agree that this is a key question that we did not adequately address in the original manuscript. One of the major goals we set during the revision process was to develop a native elongating transcript sequencing (NET-SEnd-seq) protocol for Mtb as recommended by the reviewer. After several rounds of optimization for cell lysis, RNAP pull-down, and isolation of bound RNA for SEnd-seq, we succeeded in obtaining high-quality data that specifically captured RNAP-associating transcripts. As illustrated in the new Fig. 3 and Extended Data Fig. 10, the RNA coverage from NET-SEnd-seq exhibits a similar pattern to that from total RNA SEnd-seq (i.e., a significant drop-off 200-500 nt downstream of TSS for low-PF TUs and elevated downstream signals for high-PF TUs), suggesting that most if not all short transcripts remain bound to paused RNAPs rather than released post-termination. This is further corroborated by our new Mtb RNAP ChIP-seq data, which we obtained during revision

with an improved protocol, that showed RNAP occupancy peaks overlapping with the drop-off regions from SEnd-seq (new Fig. 3 and Extended Data Fig. 11). Together, these results provide strong evidence that the short transcripts in Mtb are predominantly associated with paused elongation complexes rather than post-termination released species. The NET-SEnd-seq method is now reported in the revised manuscript (Lines 498-537). We thank the reviewer for this excellent suggestion.

3. Proteomic analysis of Linezolid treated cells. Contingent on the answer to my comment above, truncated proteins are expected to accumulate in Linezolid treated cells, when transcription-translation coupling is disturbed. Can the authors detect an enrichment of truncated proteins under any condition in *M. tuberculosis*, and not for example in *M. smegmatis* where PF values are higher at baseline?

Response: Per our answer to the comment above, our new data suggest that the short Mtb transcripts are mostly associated with paused elongation complexes instead of post-termination species. These short transcripts are also not typically associated with ribosomes as suggested by our analysis of Mtb ribosome profiling data (e.g., Fig. 5b). Once the RNAP is coupled to translating ribosomes, it tends to continue elongation and synthesize full-length mRNAs (e.g., Fig. 5a). Therefore, we do not expect a significant accumulation of truncated proteins. Proteomic analysis of Mtb cells under different conditions would still be very interesting, but we feel is outside the scope of the current work.

4. The role of sigma A in the low processivity of the transcription machinery of *M. tuberculosis*. In Figure 3 and Extended Data Figure 10, the authors present length analysis of RNA products from in vitro transcription reactions. Whenever sigma A is bound to the core enzyme, it has an adverse effect on transcription processivity. This result is especially striking with a scaffold, with a pre-formed bubble, that obviates the need for sigma. Still, it is not clear whether the in vitro results emulate the behavior of sigma in living cells. Do the authors record a different gene-PF values depending on which sigma factor is transcribing a given gene in *M. tuberculosis*? How does the closely-related *M. smegmatis* sigma factor affect transcription processivity in the same type of assay?

Response: We appreciate this comment. To connect the in vitro effect of σ^A to its in vivo behavior, we performed Mtb σ^A ChIP-seq experiments to map the binding positions of σ^A in the Mtb genome. We found that σ^A occupancy peaks closely overlap with the RNA coverage drop-off regions detected by SEnd-seq (shown in the new Fig. 3b, c and Extended Data Fig. 12), supporting our model that σ^A induces Mtb RNAP pausing 200-500 nt downstream of TSS. Under our experimental conditions, the majority of Mtb genes is expected to be controlled by the housekeeping σ^A , consistent with our earlier finding that σ^A is the only essential Mtb σ factor under these conditions (Bosch et al. PMID 34297925). The regulons of most alternative σ factors respond to diverse stresses and are typically not highly expressed under standard lab growth conditions. We generated

Mtb σ^A ChIP-seq data during the revision process, which took a significant amount of time. High-quality ChIP-seq data of the other Mtb σ factors are unfortunately not available. Nonetheless, we did perform in vitro transcription assays with purified σ^B and σ^F proteins and found that they also induced the accumulation of short transcripts. We also performed in vitro assays with the M. smeg RNAP/ σ^A holoenzyme and found that it produced mostly incomplete transcripts similar to its Mtb counterpart. These in vitro results (now shown in the new Extended Data. Fig. 14g-i) indicate that σ generally induces transcriptional pausing in mycobacteria. Elucidating how the diverse σ factors differentially influence RNAP pausing is an interesting area of future research.

5. Recent publications suggest that incomplete transcripts dominate the transcriptome of other bacteria species, including that of *E. coli*. The authors should cite the work of Herzel et al. (PMID: 35524564).

Response: We thank the reviewer for suggesting this reference. Notably, they reported that the majority of incomplete transcripts in *E. coli* are decay intermediates whereas nascent RNAs only contribute to a small fraction. In contrast, we show that the incomplete transcripts in Mtb are predominantly nascent RNAs. This distinction highlights the differential regulation of the RNA life cycle in different bacterial species. We have added a discussion on this point in the revised manuscript (Lines 222-225).

Minor

“...and Eco RNAP can retain sigma70 throughout the transcription cycle in some contexts” the authors may cite an earlier paper here as well (PMID: 11525730).

Response: We thank the reviewer for pointing out this reference. It is now cited in the revised manuscript (Line 231).

Referee #2 (Remarks to the Author):

The authors use SEnd-seq to map full-length transcripts in *Mycobacterium tuberculosis* (Mtb). They show that for most genes, the transcripts produced are frequently shorter than the gene, suggesting incomplete transcription. This is in contrast to the situation in *Escherichia coli*, where most transcripts appear to be full-length. The authors argue that Mtb RNA polymerase has lower processivity than RNAP polymerase from *E. coli*, and that this difference explains why Mtb transcripts are often shorter than the associated genes. Lastly, the authors show that impaired translation leads to production of even shorter transcripts for protein-coding genes, suggesting that transcription processivity is enhanced by coupled translation.

The observation that Mtb transcripts are often incomplete is interesting because of the potential physiological implications for translation. Unfortunately, the authors do not address the impact of pervasive premature transcription on Mtb physiology. One

important area of investigation would be the impact on tmRNA usage. With so many prematurely terminated transcripts for mRNAs, you would expect an outsized role for tmRNA in Mtb since tmRNA is involved in dealing with non-stop translation (i.e., mRNAs with a start codon but no stop codon, where otherwise ribosomes would become trapped at the 3' end). It would also be interesting to look at the impact of premature transcription termination on ORF and/or operon size, and gene position within operons.

In addition, there is insufficient evidence to conclude that RNA polymerase processivity is the cause of frequent incomplete transcripts. Additionally, the authors have not ruled out a role for Rho-dependent polarity in the reduced processivity of transcription under conditions of translation inhibition.

Assuming that the pervasiveness of incomplete transcripts is indeed due to the compromised RNA polymerase processivity, the impact of the paper could be increased by determining the mechanism that explains why Mtb RNA polymerase is less processive than its *E. coli* counterpart. The authors present some evidence that implicates the Sigma factor. This is an intriguing possibility, but there are no data presented that can explain how the Sigma factor impacts transcription elongation, and there is extensive evidence that most Sigma factor is rapidly released from elongating RNA polymerase in vivo.

Response: We appreciate the reviewer's positive evaluation and constructive critiques on our work. We have performed new experiments and analyses to address the specific points as detailed below. Here we would like to briefly respond to the reviewer's general comments. Regarding the study of tmRNA usage in Mtb, our new NET-SEnd-seq data show that the short RNAs are predominantly native elongating transcripts associated with paused RNAPs rather than prematurely terminated transcripts. When coupled with active translation, these paused transcripts could be further extended and reach the stop codon. Therefore, we do not think there is significant non-stop translation and thus did not pursue the investigation of tmRNA usage in Mtb.

We have obtained more evidence for the differences between Mtb and *E. coli* RNAPs in terms of their genomic binding and transcriptional output (see our response to major comment #1). We have evaluated the effect of translation inhibition on transcription elongation in Rho-depleted Mtb cells and did not find evidence for Rho-dependent polarity (see our response to major comment #2). We also performed additional experiments, including σ^A ChIP-seq experiments and in vitro transcription assays, to support the key contribution of σ factors to the prevalence of short transcripts in Mtb.

To our knowledge, our study is the first to describe pervasive promoter-proximal pausing of RNAP in mycobacteria. We also describe mechanisms for how such pausing can be induced (σ factor binding) and rescued (transcription-translation coupling). Our results depict a model for mycobacterial transcription that is drastically distinct from the one well established for *E. coli*. We now address the impact of our work in the revised manuscript by discussing the implications of prevalent RNAP pausing for Mtb physiology and TB therapy.

Major comments:

1. The authors use ChIP-seq of RNA polymerase to make the argument that Mtb RNA polymerase is less processive than RNA polymerase from *E. coli*. However, they do not present a comparison of Mtb RNA polymerase ChIP-seq data to *E. coli* RNA polymerase ChIP-seq data. And in fact, a previous study (PMID 19150431; not cited in the current study) showed a very similar profile of RNA polymerase ChIP-seq in *E. coli*, suggesting that processivity of *E. coli* and Mtb RNA polymerases are similar in vivo. In vitro data support the authors' claim; however, there are many caveats associated with these data that are not discussed, including the lack of elongation factors (e.g., NusA, NusG), the potential that differences in reaction conditions could impact processivity, and the fact that only one template is used to compare *E. coli* and Mtb RNA polymerases.

Response: We appreciate this comment and apologize for not making the distinctions between *E. coli* and Mtb RNAPs more clear. The *E. coli* RNAP ChIP-seq data in Mooney et al. PMID 19150431 (now cited in the revised manuscript) show peaks centered around TSS, suggesting that most RNAPs are poised at or near the promoter. Once the RNAP enters the elongation phase, it tends to transcribe the entire gene body as suggested by the lack of RNAP occupancy peaks within the transcribed region. In contrast, the Mtb RNAP ChIP-seq data that we obtained during revision show peaks 200-500 nt downstream of TSS for most TUs, suggesting that most RNAPs have escaped their promoters but paused within a defined window in the transcribed region. The RNAP peak position agrees nicely with the nascent transcript length from our NET-SEnd-seq data. To enable a more direct comparison, we presented Mtb and *E. coli* RNAP ChIP-seq data side by side in the new Extended Data Fig. 11g and h, clearly exhibiting different distributions.

We also performed additional in vitro transcription experiments to strengthen our conclusion. We measured *E. coli* and Mtb transcription on two other DNA templates using the same reaction conditions and observed the accumulation of short transcripts for Mtb RNAP/ σ^A but not for *E. coli* RNAP/ σ^{70} . We also evaluated the effect of NusA and NusG using purified Mtb proteins and did not observe a significant change by either factor under these conditions. These results are now shown in the new Extended Data Fig. 14.

2. Figure 4 looks at the effect of translation inhibition on transcription processivity. The authors conclude that inhibiting translation causes a reduction in transcription processivity on mRNAs, independent of Rho. However, they do not test whether depleting Rho reverses the effect of translation inhibition on transcription processivity. Hence, there are insufficient data to conclude that the effect of translation inhibition on transcription processivity is independent of Rho.

Response: We thank the reviewer for raising this important point. As suggested, we performed new experiments to evaluate the effect of translation inhibition on transcription elongation in Rho-depleted Mtb cells. Successful Rho depletion was confirmed at both RNA and protein levels (Extended Data Fig. 13). We found that Rho depletion did not reverse the reduction in PF upon linezolid treatment (now presented in the new Fig. 5d). This result is consistent with our model that Mtb RNAP is paused, not terminated, when

uncoupled from translation, as Rho-mediated termination does not seem to play a significant role in transcription-translation coupling in *Mtb*, unlike in *E. coli* where Rho is required for the polarity effect. The result and discussion on these experiments have been added to the revised manuscript (Lines 206-207 and 243-248).

Additional comments:

1. The authors describe two different SEnd-seq methods in the methods section. One method enriches for primary transcripts while the other does not. The authors do not indicate which method was used for the data presented in the paper. This is particularly important since it impacts how some of the data are interpreted. For example, Figure 1b shows many short transcripts associated with the Rv0517 gene that the authors attribute to premature transcription termination. If these data are from primary transcripts only, it is possible that there are other (processed) transcripts whose 5' ends align with the 3' ends of the short transcripts that initiate at the TSS. Such a result would strongly suggest a role for RNase processing. Assuming the authors have data for both methods, they should show the data side by side.

Response: We apologize for this confusion. To clarify, we used the primary RNA SEnd-seq data to identify TSSs and the total RNA SEnd-seq data for other analyses including the calculation of PF values. This is now clearly stated in the text. The data shown in the original Fig. 1b is total RNA data. In the new Fig. 1b and Extended Data Fig. 2a, b, 4a, b, we have included a side-by-side comparison of primary RNA and total RNA data, both of which show a similar prevalence of short transcripts. We rarely detected in the total RNA dataset processed transcripts with their 5' ends aligning with the 3' ends of short transcripts initiating at TSS.

2. The authors show that inducing expression of *lacZ* transcription in *Mtb* leads to accumulation of RNA primarily at the 5' end of the gene. They argue that these data support a model in which short transcripts are generated due to premature transcription termination rather than RNA degradation, but the logic for this argument is not clearly explained.

Response: The rationale for this experiment is as follows. Because we can induce *lacZ* transcription in a controlled manner, we were able to track the progression of the RNA length profile as a function of post-induction time, as opposed to the steady-state profile from the SEnd-seq data. If there were significant RNase-mediated processing that gives rise to the short transcripts, we would still be able to observe increasing RNA signals in the downstream region, especially at early time points (along the same line as the reviewer's previous comment). However, we only observed the accumulation of incomplete *lacZ* transcripts that were 200-500 nt in length, indicating that they were not generated by RNA processing. We have added an explanation for the rationale of this experiment to the revised manuscript (Lines 110-115). Furthermore, we knocked down

individual RNase or RNase-related genes in Mtb using CRISPRi and did not find any consistent increase in the PF value for the tested TUs (new Extended Data Fig. 9). These results suggest that the short RNAs are most likely generated during nascent transcription rather than post-transcriptional processing. We note that, in light of our new NET-SEnd-seq and ChIP-seq data, we think that the short transcripts are mostly associated with paused RNAPs rather than released upon premature termination.

3. SEnd-seq is a good approach to tackle the questions posed here, but RNA-seq would also suffice to determine whether there is widespread premature transcription termination. The authors should compare their SEnd-seq data to some of the many published RNA-seq datasets for Mtb. If the authors' model is correct, you would expect to see a similar drop-off in RNA-seq reads through transcribed regions as seen with SEnd-seq.

Response: We agree with the reviewer that a similar RNA coverage drop-off through transcribed regions should also be observed from standard RNA-seq datasets. To this end, we analyzed a previously published standard RNA-seq dataset for Mtb (Aguilar-Ayala et al. PMID 29247215) using the same analysis pipeline as we did for SEnd-seq. The results, presented in the new Extended Data Fig. 7, indeed show a drop-off pattern that matches the SEnd-seq results. The SEnd-seq profile shows more pronounced boundary features because of its higher resolution in mapping transcript termini, a major advantage of the SEnd-seq method. Nevertheless, the overall RNA expression levels are comparable between the two methods. We thank the reviewer for this great suggestion.

4. ChIP-seq of RNA polymerase is a reasonable approach for mapping the distribution of RNA polymerase across a transcribed region. However, it cannot distinguish between signal from initiating RNA polymerase and signal from elongating RNA polymerase. NET-seq would be a better approach since it looks only at elongating RNAP, and it provides single nucleotide resolution.

Response: We thank the reviewer for this excellent suggestion. We spent significant efforts during the revision process developing a protocol for Mtb NET-seq, which has not been published before. After several rounds of optimization for cell lysis, RNAP pull-down, and nascent RNA isolation, we were able to obtain high-quality NET-SEnd-seq data reporting simultaneously the 5' and 3' ends of nascent elongating transcripts. Importantly, the NET-SEnd-seq data match well with the Mtb RNAP ChIP-seq data, i.e., the RNAP occupancy peaks overlap with the 3' ends of the nascent transcripts, both in individual TUs and in the genome-wide average, suggesting that the short transcripts are mostly associated with paused transcription elongation complexes. These results are now shown in the new Fig. 3 and Extended Data Fig. 10, 11. The NET-SEnd-seq method is also described in the revised manuscript (Lines 498-537).

5. The authors compare their Mtb SEnd-seq data to E. coli data, but they do not mention

the source of the *E. coli* data. Are the *E. coli* data the ones previously published in reference 3?

Response: Yes. The *E. coli* SEnd-seq data are from our previous publication (Ju et al. 2019 PMID 31308523), which is now referenced in the text where we make the comparison (Line 79).

6. The authors should compare the TSSs they identify to those identified by two earlier studies (reference 8 and PMID 26536359). They could also look to see if the putative TSSs have the expected promoter sequences upstream. This would increase confidence in the TSS assignments.

Response: We thank the reviewer for this suggestion. We have performed a comparative analysis between the TSSs that we identified using SEnd-seq and those identified in Cortes et al. (PMID 24268774) and Shell et al. (PMID 26536359). Our dataset recapitulates the vast majority of previously annotated TSSs, but also reveals many new sites. We also analyzed the sequences upstream of our identified TSSs, which yielded the conserved -10 motif, adding confidence to the TSS assignments. These results are now shown in the new Extended Data Fig. 2f-i.

7. The authors assess the position of *Mtb* TSSs relative to genes and then compare these data to data for other bacterial species. They conclude that *Mtb* has more antisense RNAs than other species. I would be hesitant draw a strong conclusion given that small technical differences can have an outsized impact on TSS identification (see PMID 25266388).

Response: We appreciate this comment and acknowledge that TSS identification is sensitive to the choice of technical parameters. We used the same criteria for TSS identification in all the bacterial species that we profiled by SEnd-seq. Nevertheless, as the reviewer suggested, we have refrained from making a strong conclusion about the difference between *Mtb* and *E. coli* in terms of their TSS distributions and have modified the figure accordingly (new Extended Data Fig. 2c).

8. The authors should compare the list of 747 putative leaderless ORFs they identify based on SEnd-seq data to leaderless ORFs identified in earlier studies ((reference 8 and PMID 26536359). The example they show in Extended Data Figure 2f has been described previously (reference 8) as a leaderless ORF.

Response: We thank the reviewer for this suggestion. We have performed a comparative analysis between our set of leaderless TSSs and those identified in the two earlier studies. The results are now shown in the new Extended Data Fig. 2g. We have also provided an alternative example of leaderless ORF that has not been reported before (new Extended Data Fig. 2e).

9. The authors should mention that a previous study (reference 8) also detected extensive antisense transcription in *Mtb*.

Response: We apologize for this omission. This reference is now cited in the revised manuscript where we describe the antisense transcript results (Line 70).

10. The authors should compare their SEnd-seq data from cells with depleted Rho to RNA-seq data in reference 13. There appear to be some substantial differences in the datasets between the two studies; data in reference 13 are consistent with transcripts being extended in cells with depleted Rho.

Response: We thank the reviewer for this suggestion. We have comprehensively analyzed the RNA-seq data in Botella et al. and compared them to our SEnd-seq data. In fact, the two studies are remarkably consistent with each other. First, even though the RNA-seq results showed less pronounced features of the RNA boundaries than SEnd-seq as expected from their methodological differences, both datasets showed a similar drop-off in RNA coverage within 500 nt downstream of TSS, and that Rho depletion did not significantly change the drop-off pattern. There may be some cases where Rho depletion resulted in extended transcripts as reported in Botella et al., but it did not cause a global increase in transcript length. Second, both studies found that Rho depletion caused a genome-wide increase in the abundance of antisense transcripts. We have described these comparisons in the new Extended Data Fig. 13h, i and main text of the revised manuscript (Lines 158-161).

11. In the Discussion, the authors hypothesize that translation can "push" transcription, increasing processivity. They suggest that this phenomenon does not occur in *E. coli*. However, reference 14 provides evidence for exactly this mechanism in *E. coli*.

Response: We apologize for causing this confusion. We did not intend to suggest that the physical coupling between transcription and translation does not occur in *E. coli*, as reference 14 has provided clear evidence for it. We intended to highlight the difference between *E. coli* and *Mtb* in terms of the involvement of Rho in their transcription-translation coupling. In *E. coli*, Rho is required for the polarity effect by mediating premature termination when RNAP is uncoupled from the ribosome. On the other hand, *Mtb* Rho depletion does not reverse the shortening of transcript length caused by translation inhibition. This point is now clarified in the revised manuscript (Lines 243-248).

Reviewer Reports on the First Revision:

Referees' comments:

Referee #1 (Remarks to the Author):

The authors have diligently addressed all of my previous comments. I would like to propose a few additional suggestions for them to consider when finalizing the manuscript:

1. I commend the authors for developing NET alongside the SEnd sequencing protocol. While this topic may lie outside the scope of this manuscript, it might be worth mentioning, possibly in the discussion section, how this new protocol could facilitate the stratification of transcription elongation complexes based on sigma occupancy. Such an analysis could further solidify the role of sigma in RNAP pausing in *M. tuberculosis*. For reference, please see PMID: 33568644.
2. Could the authors attempt to derive a consensus pause sequence from the NET data combined with SEnd sequencing data, particularly at the primary pause site within each transcription unit? This effort might help shed light on the broader question of which signals or sequence features contribute to RNAP pausing within the 200-500 nucleotide window in *M. tb*.
3. In the revised version, the authors did not present data supporting a regulatory function for pervasive pausing within the 200-500 nucleotide window from the TSS. Therefore, I suggest that the authors briefly consider the possibility that such a regulatory role might not exist and that the purpose of pausing could be different.

Additional comments to provisional rebuttal to comments 1-3 by referee #2:

After carefully reviewing the author's rebuttal to the reviewer's comments, I believe that the model proposed by the authors is well-supported by their data.

First, the NET-SEnd results clearly indicate that incomplete transcripts are indeed associated with RNAP. Distinguishing whether all paused complexes resume transcription elongation or not is a challenging task, given that RNAP pausing is also an integral step of intrinsic transcription termination. Unless the reviewer has a specific experiment in mind that could unequivocally differentiate between the two, I find it reasonable for the authors to address this question in the manuscript's discussion. I don't believe a new round of experiments is necessary.

Second, the meta-analysis of RNAP ChIP-seq data from rifampicin-treated cells effectively illustrates that RNAP is trapped at the promoters of highly expressed genes. The prominence of this effect is expected and not a novel observation.

Finally, the SigA ChIP-seq data prepared by the authors should address the last comment from the reviewer.

I would like to reiterate my opinion that the manuscript is acceptable.

Referee #2 (Remarks to the Author):

The authors have substantially revised the paper. The two major additions are new ChIP-seq data for core RNAP and SigA, and NET-SEnd-seq data that map the position of elongating RNAP genome-wide. These new data are accompanied by an expanded model in which RNAP frequently pauses 200-500 nt into transcripts until engaged by an upstream ribosome that promotes further transcription elongation. The NET-SEnd-seq data are an important addition to the paper. While there is an additional analysis that would increase confidence in the data, assuming the NET-SEnd-seq data are good, they argue strongly against short transcripts being generated by post-transcriptional processing, and instead strongly implicate the RNAP. This is an important addition to the work. However, I am unconvinced by the authors' expanded model. The NET-SEnd-seq data provide no direct evidence for pausing; they are more consistent with stochastic transcription termination early in transcripts. Moreover, the new ChIP-seq data are inconsistent with similar data from other groups. In particular, the new SigA ChIP-seq data do not show the expected result (i.e., that SigA ChIP-seq signal peaks very close to TSSs), and differ from previous studies that do show the expected result. Overall, the new data increase my confidence that that the phenomenon the authors observe is due to a property of the RNAP rather than a post-transcriptional processing event, but the mechanism remains unclear.

Major comments:

1. The authors argue that short transcripts are associated with paused elongation complexes. The NET-seq data support the idea that the short transcripts are RNAP-associated, as opposed to being the product of ribonuclease activity on completed transcripts, but I don't see evidence that the transcripts are associated with paused RNAPs. Pause sites would be indicated by "spikes" in the NET-SEnd-seq data (see e.g., PMIDs 24789973 and 24926020). Alternatively, if RNAP pauses stochastically and without sequence specificity, there would be an overall increase in NET-SEnd-seq signal in the pausing region, which there is not. The NET-SEnd-seq data are more consistent with stochastic transcription termination early in transcripts.
2. The new ChIP-seq data for SigA differ considerably from published data from other groups. Specifically, the "peak" of SigA coverage, on average, is >100 bp downstream of TSSs (Figure 3c). This is in contrast to previous ChIP-seq studies of SigA in mycobacteria (e.g., PMID 25089258, <https://www.biorxiv.org/content/10.1101/2023.03.16.533064v1>) that show the SigA peak roughly at the TSS. Numerous ChIP-seq studies of Sigma factors in other bacterial species all show the peak roughly at TSSs, consistent with the large majority of Sigma factor being rapidly released from elongating RNAP. It is formally possible that Mtb RNAP is different, and retains SigA throughout the transcription cycle, but this is inconsistent with previous data from mycobacteria, and would suggest a very different mechanism of transcription elongation in Mtb as compared to other bacterial species.
3. ChIP-seq data for rifampicin-treated cells are also very different to the expectation. Specifically, you would expect to see RNAP "trapped" at promoters, as you do in *E. coli*, but instead data in

Supplementary Figure 11f show “flat” RNAP signal.

Additional comments:

1. The new NET-SEnd-seq data strengthen the argument that the abundance of short transcripts is due to a property of the RNAP rather than post-transcriptional processing. However, because this is a new method, and this is the first time NET-seq has been applied in a genome with a high G/C content, I think some additional analysis is needed to conclude that NET-SEnd-seq identifies full-length nascent transcripts. Specifically, the authors should observe an “elemental” pause motif associated with predominant 3’ ends.
2. The authors’ model proposes that RNAP pauses 200-500 nt into a transcript prior to translation initiation. Given what we know about Rho, including Rho in Mtb, it is very unlikely that RNAP could transcribe 200-500 nt untranslated RNA without Rho termination. In the absence of another model, this warrants some discussion.
3. The authors show NET-SEnd-seq data for only a few individual loci, and the plots are difficult to interpret because it appears that at some positions reads pile up beyond the scale shown in the figure. It would be more informative to plot the abundance of RNA 3’ end positions for sequences whose 5’ end corresponds to the TSS. This would make the presence and strength of pause sites more obvious, although I see little evidence of such pausing in the data already presented.

Author Rebuttals to First Revision:

Point-to-point response to referees' comments:

Referee #1 (Remarks to the Author):

The authors have diligently addressed all of my previous comments. I would like to propose a few additional suggestions for them to consider when finalizing the manuscript:

Response: We thank the reviewer for these additional suggestions and have incorporated them in the revised manuscript.

1. I commend the authors for developing NET alongside the SEnd sequencing protocol. While this topic may lie outside the scope of this manuscript, it might be worth mentioning, possibly in the discussion section, how this new protocol could facilitate the stratification of transcription elongation complexes based on sigma occupancy. Such an analysis could further solidify the role of sigma in RNAP pausing in *M. tuberculosis*. For reference, please see PMID: 33568644.

Response: We agree with the reviewer that developing a NET-SEnd-seq protocol using His-tagged sigma factors would certainly be valuable for dissecting sigma occupancy in paused elongation complexes and solidifying the role of sigma in *Mtb* RNAP pausing. We have added this point as a future direction as well as the reference in the discussion section (Lines 231-232).

2. Could the authors attempt to derive a consensus pause sequence from the NET data combined with SEnd sequencing data, particularly at the primary pause site within each transcription unit? This effort might help shed light on the broader question of which signals or sequence features contribute to RNAP pausing within the 200-500 nucleotide window in *M. tb*.

Response: We appreciate this comment. We have analyzed the sequences around the pausing sites defined by the 3' ends in the NET-SEnd-seq data. As shown in **Figure R1**, we didn't find significant sequence features using various threshold values in the analysis (higher reads correspond to stronger pauses). Only with the highest threshold value did we observe minor enrichments downstream of the pausing site, but only a small number of sites were qualified. These results contrast with the consensus pause sequence observed in *E. coli* (e.g., PMID: 24789973). Thus, the prevalent *Mtb* RNAP pausing in the 200-500

nucleotide window downstream of TSS does not appear to be driven by sequence features, instead is likely mediated by external factors such as sigma. This result is now shown as Extended Data Fig. 7n in the revised manuscript.

Figure R1. Motif analysis of DNA sequences surrounding the Mtb RNAP pause sites defined by the RNA 3' ends in the NET-SEnd-seq data. Each panel uses a different read threshold. Positions with higher reads indicate stronger pause sites.

3. In the revised version, the authors did not present data supporting a regulatory function for pervasive pausing within the 200-500 nucleotide window from the TSS. Therefore, I suggest that the authors briefly consider the possibility that such a regulatory role might not exist and that the purpose of pausing could be different.

Response: Although we made speculations based on what has been learned from other organisms, we acknowledge that the regulatory function for the pervasive transcriptional pausing in Mtb, if any, may be different and requires further investigation. We have now mentioned this caveat in the discussion section (Lines 253-254).

Additional comments to provisional rebuttal to comments 1-3 by referee #2:

After carefully reviewing the author's rebuttal to the reviewer's comments, I believe that the model proposed by the authors is well-supported by their data.

First, the NET-SEnd results clearly indicate that incomplete transcripts are indeed associated with RNAP. Distinguishing whether all paused complexes resume transcription elongation or not is a challenging task, given that RNAP pausing is also an integral step of intrinsic transcription termination. Unless the reviewer has a specific experiment in mind that could unequivocally differentiate between the two, I find it reasonable for the authors to address this question in the manuscript's discussion. I don't believe a new round of experiments is necessary.

Second, the meta-analysis of RNAP CHIP-seq data from rifampicin-treated cells effectively illustrates that RNAP is trapped at the promoters of highly expressed genes. The prominence of this effect is expected and not a novel observation.

Finally, the SigA CHIP-seq data prepared by the authors should address the last comment from the reviewer.

I would like to reiterate my opinion that the manuscript is acceptable.

Response: We appreciate the reviewer's assessment that our model is supported by the data and that the manuscript is acceptable. To further address referee #2's remaining concerns, we have collected new RNAP CHIP-seq data from rifampicin-treated cells and SigA CHIP-seq data from both Mtb and Msm, which are described in detail below.

Referee #2 (Remarks to the Author):

The authors have substantially revised the paper. The two major additions are new ChIP-seq data for core RNAP and SigA, and NET-SEnd-seq data that map the position of elongating RNAP genome-wide. These new data are accompanied by an expanded model in which RNAP frequently pauses 200-500 nt into transcripts until engaged by an upstream ribosome that promotes further transcription elongation. The NET-SEnd-seq data are an important addition to the paper. While there is an additional analysis that would increase confidence in the data, assuming the NET-SEnd-seq data are good, they argue strongly against short transcripts being generated by post-transcriptional processing, and instead strongly implicate the RNAP. This is an important addition to the work. However, I am unconvinced by the authors' expanded model. The NET-SEnd-seq data provide no direct evidence for pausing; they are more consistent with stochastic transcription termination early in transcripts. Moreover, the new ChIP-seq data are inconsistent with similar data from other groups. In particular, the new SigA ChIP-seq data do not show the expected result (i.e., that SigA ChIP-seq signal peaks very close to TSSs), and differ from previous studies that do show the expected result. Overall, the new data increase my confidence that that the phenomenon the authors observe is due to a property of the RNAP rather than a post-transcriptional processing event, but the mechanism remains unclear.

Response: We appreciate this reviewer's increased confidence in our revised manuscript. Below we clarify our NET-SEnd-seq data to show that they indeed support transcriptional pausing. We also performed new SigA ChIP-seq experiments in both Mtb and Msm and show that there is no discrepancy between our data and previously published results.

Major comments:

1. The authors argue that short transcripts are associated with paused elongation complexes. The NET-seq data support the idea that the short transcripts are RNAP-associated, as opposed to being the product of ribonuclease activity on completed transcripts, but I don't see evidence that the transcripts are associated with paused RNAPs. Pause sites would be indicated by "spikes" in the NET-SEnd-seq data (see e.g., PMIDs 24789973 and 24926020). Alternatively, if RNAP pauses stochastically and without sequence specificity, there would be an overall increase in NET-SEnd-seq signal in the pausing region, which there is not. The NET-SEnd-seq data are more consistent with stochastic transcription termination early in transcripts.

Response: We apologize for not adequately explaining our NET-SEnd-seq data. Unlike previous NET-seq data in the cited references (PMIDs 24789973 and 24926020) where the 3' ends of nascent transcripts were plotted, our NET-SEnd-seq data plot the full-length sequences of nascent RNA from 5' end to 3' end. When we just plot the 3' ends, we did observe a peak in the 200-300 nt region downstream of TSS (**Figure R2**), consistent with RNAP pausing in this region. If transcription were terminated rather than paused, we

wouldn't expect to detect those transcripts in the NET-SEnd-seq sample because terminated transcripts will dissociate from RNAP. This plot is now shown as Fig. 3b in the revised manuscript.

Figure R2. Summed intensities of the 5'-end position (green) and 3'-end position (red) of nascent RNAs from the NET-SEnd-seq dataset ($n = 4,362$).

We also performed DNA sequence analysis for the 3' ends of nascent transcripts and did not find any noticeable motif (see **Figure R1** above in our response to Reviewer #1), indicating that RNAP pausing in this region is not sequence specific.

2. The new ChIP-seq data for SigA differ considerably from published data from other groups. Specifically, the “peak” of SigA coverage, on average, is >100 bp downstream of TSSs (Figure 3c). This is in contrast to previous ChIP-seq studies of SigA in mycobacteria (e.g., PMID 25089258, <https://www.biorxiv.org/content/10.1101/2023.03.16.533064v1>) that show the SigA peak roughly at the TSS. Numerous ChIP-seq studies of Sigma factors in other bacterial species all show the peak roughly at TSSs, consistent with the large majority of Sigma factor being rapidly released from elongating RNAP. It is formally possible that Mtb RNAP is different, and retains SigA throughout the transcription cycle, but this is inconsistent with previous data from mycobacteria, and would suggest a very different mechanism of transcription elongation in Mtb as compared to other bacterial species.

Response: We appreciate this comment and acknowledge that our Mtb SigA ChIP-seq pattern is different from those previously reported for other bacterial species. To rule out the possibility that this disparity originates from an inherent problem with our ChIP-seq protocol, we used the same protocol to conduct SigA ChIP-seq experiments in *M. smegmatis* (Msm). We observed very similar SigA occupancy peaks to previous Msm data (PMID: 25089258), i.e., they were mostly located at the TSS (**Figure R3**). Therefore, we believe that the unusual SigA ChIP-seq pattern reflects a unique mechanism of transcription elongation

in *Mtb*, which may be mediated by the extended N-terminal disordered region in the *Mtb* SigA or distinct transcription kinetics associated with *Mtb*'s slow growth.

3. ChIP-seq data for rifampicin-treated cells are also very different to the expectation. Specifically, you would expect to see RNAP “trapped” at promoters, as you do in *E. coli*, but instead data in Supplementary Figure 11f show “flat” RNAP signal.

Response: We appreciate this comment and examined our data in more detail. For the *Mtb* transcription units that still showed relatively strong RNAP occupancy in the rifampicin-treated condition ($n = 65$), the RNAP ChIP-seq peak indeed shows a shift from the downstream of TSS to the promoter region upon rifampicin treatment (**Figure R4**). We have since repeated these experiments and obtained similar results. These results are now shown as Extended Data Fig. 7d-f in the revised manuscript. Moreover, we also treated *Msm* cells with rifampicin and performed RNAP ChIP-seq experiments (**Figure R5**). A similar shift from the TSS downstream region to the promoter region was observed for the *Msm* TUs with strong RNAP occupancy in the rifampicin-treated condition ($n = 445$). Based on these reproducible results, we are confident with our interpretation of the data. We speculate that the RNAPs at weak promoters are refractory to efficient crosslinking and detection by ChIP-seq, which gives rise to the “flat” signal.

Figure R4. a, Mtb RNAP ChIP-seq and total RNA SEnd-seq data tracks for two example genomic regions from cells treated with DMSO or rifampicin. **b**, Summed RNAP ChIP-seq intensities for Mtb TUs under the DMSO- or rifampicin-treated condition aligned at TSSs.

Figure R5. a, Msm RNAP ChIP-seq data tracks for two example genomic regions from cells treated with DMSO or rifampicin. **b**, Summed RNAP ChIP-seq intensities for Msm TUs under the DMSO- or rifampicin-treated condition aligned at TSSs.

Additional comments:

1. The new NET-SEnd-seq data strengthen the argument that the abundance of short transcripts is due to a property of the RNAP rather than post-transcriptional processing. However, because this is a new method, and this is the first time NET-seq has been applied in a genome with a high G/C content, I think some additional analysis is needed to conclude that NET-SEnd-seq identifies full-length nascent

transcripts. Specifically, the authors should observe an “elemental” pause motif associated with predominant 3’ ends.

Response: We appreciate this comment. The validity of the NET-SEnd-seq method has been thoroughly evaluated in the manuscript (Extended Data Fig. 6). As suggested, we have also conducted additional analysis on the sequences around the pausing sites defined by the 3’ ends in the NET-SEnd-seq data. As shown in our response to referee #1 above (**Figure R1**), we didn’t find noticeable sequence features that would suggest the existence of a consensus elemental pause motif in Mtb. Thus, the prevalent pausing in the 200-500 nucleotide window downstream of TSS does not appear to be induced by DNA sequence, again indicating a distinct regulatory mechanism for transcription elongation in Mtb.

2. The authors’ model proposes that RNAP pauses 200-500 nt into a transcript prior to translation initiation. Given what we know about Rho, including Rho in Mtb, it is very unlikely that RNAP could transcribe 200-500 nt untranslated RNA without Rho termination. In the absence of another model, this warrants some discussion.

Response: The function of Rho in Mtb transcription was investigated in our study (Extended Data Fig. 8). Our data show that Rho depletion did not change the effect of translation inhibition on transcript lengths (Fig. 5d), indicating that other factors are involved in transcription-translation coupling in Mtb and prevent Rho from directly targeting the elongation complex. This is now discussed in the revised manuscript (Lines 242-245).

3. The authors show NET-SEnd-seq data for only a few individual loci, and the plots are difficult to interpret because it appears that at some positions reads pile up beyond the scale shown in the figure. It would be more informative to plot the abundance of RNA 3’ end positions for sequences whose 5’ end corresponds to the TSS. This would make the presence and strength of pause sites more obvious, although I see little evidence of such pausing in the data already presented.

Response: As the reviewer suggested, we have plotted both 5’ end and 3’ end positions of Mtb nascent RNA from the NET-SEnd-seq data (**Figure R2**), now shown as Fig. 3b in the revised manuscript. The vast majority of 5’ ends align at TSS, whereas the 3’ ends show a broad pausing zone downstream of TSS. We have clarified the evidence for pausing in our response to the major comment #1.

Reviewer Reports on the Second Revision:

Referees' comments:

Referee #1 (Remarks to the Author):

The authors have thoroughly addressed all the remaining points. I believe the paper is now acceptable.